# APC-RL: Exceeding data-driven behavior priors with adaptive policy composition

**Finn Rietz**[*]
Department of Computer Science
Örebro University
Fakultetsgatan 1, Örebro, Sweden
finn.rietz@oru.se

**Pedro Zuidberg dos Martires**
Department of Computer Science
Örebro University
Fakultetsgatan 1, Örebro, Sweden
pedro.zuidberg-dos-martires@oru.se

**Johannes Andreas Stork**
Department of Computer Science
Örebro University
Fakultetsgatan 1, Örebro, Sweden
johannesandreas.stork@oru.se

## Abstract

Incorporating demonstration data into reinforcement learning (RL) can greatly accelerate learning, but existing approaches often assume demonstrations are optimal and fully aligned with the target task. In practice, demonstrations are frequently sparse, suboptimal, or misaligned, which can degrade performance when these demonstrations are integrated into RL. We propose Adaptive Policy Composition (APC), a hierarchical model that adaptively composes multiple data-driven Normalizing Flow (NF) priors. Instead of enforcing strict adherence to the priors, APC estimates each prior's applicability to the target task while leveraging them for exploration. Moreover, APC either refines useful priors, or sidesteps misaligned ones when necessary to optimize downstream reward. Across diverse benchmarks, APC accelerates learning when demonstrations are aligned, remains robust under severe misalignment, and leverages suboptimal demonstrations to bootstrap exploration while avoiding performance degradation caused by overly strict adherence to suboptimal demonstrations.

## 1 Introduction

Demonstration data are often used to make reinforcement learning (RL) (Sutton & Barto, 2018) feasible in challenging tasks. Examples are regularizing the policy to imitate demonstrated actions (Lu et al., 2023; Zhu et al., 2018; Hester et al., 2018) and generative modeling of demonstrated actions or action sequences (Pertsch et al., 2021; Yang et al., 2022; Singh et al., 2021). However, these approaches often make implicit but crucial assumptions about demonstrations, e.g. complete coverage of the state space and optimality, which is unrealistic in many practical settings. In fact, strictly adhering to the data even when the demonstrations are suboptimal or sparse is the main reason for failure when the dataset and the online RL task are *misaligned* (i.e., in some way sub-optimal) Dong et al. (2025); Kong et al. (2024); Zhang et al. (2023). To avoid this, we should therefore only utilize demonstration data in reinforcement learning *where* and *as long as* it is pertinent for the online RL task.

In this paper, we present Adaptive Policy Composition (APC), a novel and flexible method of using demonstration data in RL, that does not rely on optimal and complete demonstrations. In APC, we decide based on online feedback *where* and *how long* to rely on the data, instead of always adhering to the demonstrations. We propose a hierarchical RL approach consisting of a *higher-level selector* that decides between *several* actors on the lower level. Importantly, there are always exactly one *prior-free* and at least one data-driven, *prior-based* actor on the lower level. Using several prior-based

---

[*]Corresponding Author

actors is possible and allows us to include several distinct demonstration datasets that contain various behaviors. Our prior-based actors pre-train their own behavior prior with their demonstrations while the prior-free actor learns from scratch. Unlike previous approaches (Pertsch et al., 2021; Yang et al., 2022; Singh et al., 2021), the inclusion of the prior-free actor provides APC with complete flexibility to diverge from demonstrations if necessary to optimize the online RL task, for example, when no demonstrations apply. Our approach allows the use of standard (off-policy) learning algorithms to optimize these actors.

Our contributions are threefold: (i) Algorithmic: We introduce APC, a novel compositional policy architecture that combines prior-based and prior-free actors under an adaptive selector. We further propose two key mechanisms—a reward-sharing scheme that enables data-efficient training across actors, and a learning-free arbitrator selector that mitigates primacy bias—both of which are crucial for robust and efficient performance. (ii) Empirical: We demonstrate that APC achieves strong robustness under demonstration misalignment, consistently outperforming prior methods such as PARROT and imitation learning baselines. (iii) Analytical: Through ablations, we identify the selector design and reward sharing as critical components for enabling stable and efficient exploration.

## 2 BACKGROUND

### 2.1 REINFORCEMENT LEARNING

RL problems are formalized as Markov Decision Processes (MDPs). A MDP is a tuple $\mathcal{M} \equiv \langle \mathcal{S}, \mathcal{A}, r, \rho, \gamma \rangle$, where $\mathcal{S}$ is the state space, $\mathcal{A}$ is the action space, $r : \mathcal{S} \times \mathcal{A} \to \mathbb{R}$ is a reward function, $\rho : \mathcal{S} \times \mathcal{A} \to \mathcal{S}$ denotes the discrete-time state-transition-kernel, and $\gamma \in (0, 1]$ is a discount factor. The goal in RL is to find a policy $\pi : \mathcal{S} \times \mathcal{A} \to [0, 1]$ that maximizes the discounted return objective

$$J(\pi) = \mathbb{E}_{(\tau \sim \pi)} \left[ \sum_{t=0}^{\infty} \gamma^t r(\mathbf{s}_t, \mathbf{a}_t) \right], \tag{1}$$

where $\mathbf{s}_t \in \mathcal{S}$, $\mathbf{a}_t \in \mathcal{A}$, and $(\tau \sim \pi)$ is a shorthand for denoting trajectories with actions sampled form the policy $\pi$ and the state evolving according to $\rho$. RL algorithms optimize the objective in equation 1 to identify the optimal policy $\pi^* = \max_\pi J(\pi)$ which inherently requires a good exploration strategy, as well as balancing the exploration versus exploitation trade-off. Uninformed exploration such as $\varepsilon$-greedy (Sutton & Barto, 2018) or temporally extended Brownian motion (Uhlenbeck & Ornstein, 1930) appeal due to their simplicity but often fall short in complex, long-horizon, sparse-reward MDPs. Exploration based on prior knowledge, e.g., demonstrations, can be more useful in such cases, which motivates our choice of composing multiple data-driven behavior priors.

### 2.2 DATA-DRIVEN POLICY PRIORS WITH GENERATIVE LATENT SPACE MODELING

Generative models of demonstrated actions can implement effective data-driven priors for RL policies that learn in their latent space (Pertsch et al., 2021; Yang et al., 2022; Singh et al., 2021). In this paper, we follow the PARROT (Singh et al., 2021) approach and learn a state-conditioned Normalizing Flow (NF) (Rezende & Mohamed, 2015; Papamakarios et al., 2021) model $T(\mathbf{z}; \mathbf{s}) = \mathbf{a}$ with parameters $\phi$. The NF maps latent actions $\mathbf{z} \in \mathcal{Z}$ from the base distribution $\mathcal{N}(\mathbf{z}; \mathbf{0}, \mathbf{I})$ to the complex, multi-modal, per-state action distribution of the demonstration dataset $\mathcal{D} = (\mathbf{s}_i, \mathbf{a}_i)_{i=1}^N$. The NF is learned with likelihood maximization and matches the dataset distribution with the generative distribution

$$\mathcal{L}(\phi) = -\mathbb{E}_{(\mathbf{a},\mathbf{s}) \sim \mathcal{D}} \left[ \log \mathcal{N}(\tilde{T}(\mathbf{a}; \mathbf{s}, \phi); \mathbf{0}, \mathbf{I}) + \log |\det J_{\tilde{T}}(\mathbf{a}; \mathbf{s})| \right] + \text{const.}, \tag{2}$$

where $\tilde{T}(\mathbf{a}; \mathbf{s}, \phi) = T^{-1}(\mathbf{a}; \mathbf{s}, \phi) = \mathbf{z}$ is the inverse NF and $J$ is the Jacobian (from the change of variables theorem). The NF $T$ serves as a data-driven prior for RL by learning a policy in the NF's latent space $\mathcal{Z}$. Specifically, a *latent policy* $\pi_{\mathbf{z}} : \mathcal{S} \times \mathcal{Z} \to [0, 1]$ outputs a latent action $\mathbf{z}$, which is then deterministically mapped to an MDP action $T(\mathbf{z}; \mathbf{s}) = \mathbf{a} \in \mathcal{A}$ through the NF transformation. We refer to the resulting policy distribution in the MDP action space as the *prior-based actor* $\pi_{\mathbf{a}}(\mathbf{a} \mid \mathbf{s})$.

RL in the latent action space $\mathcal{Z}$ of a pre-trained NF is beneficial because it focuses learning and exploration on useful, demonstrated actions in $\mathcal{A}$. Moreover, NFs are multimodal and allow a simple, unimodal latent policy $\pi_{\mathbf{z}}$ to induce a multimodal distribution in the action space. Finally, unlike

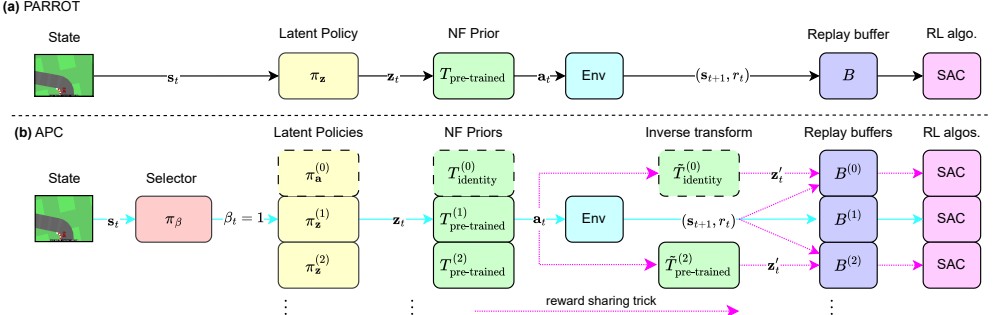

Figure 1: Architecture overview. **(a)**: PARROT (Singh et al., 2021) features a single latent policy and NF prior. **(b)**: Our method uses a high-level selector to compose multiple latent policies and NF priors. A prior-free actor (index $(0)$, dashed border) learns directly in the action space. The selected latent policy and NF prior (cyan-colored arrows) are executed at time $t$. A reward-sharing trick (magenta-colored arrows) allows us to compute the latent coordinate $\mathbf{z}'_t$ corresponding to the executed action $\mathbf{a}_t$, and to use the transition at time $t$ to also update the other actors that were not selected.

other latent space generative models used as data-driven priors in RL such as VAEs (Pertsch et al., 2021; Yang et al., 2022), NFs are invertible. This means that the prior-based actor, $\pi_{\mathbf{a}}(\mathbf{a} \mid \mathbf{s})$, in theory, can undo the behavior prior to learn any desired policy. However, in practice this has been found to be infeasible, as reported by Singh et al. (2021) and confirmed in our evaluation, leading to permanent influence of the behavior prior and resulting in failure under misalignment. This serves as the primary motivation for our method, which we present in the following section.

## 3 METHOD: ADAPTIVE POLICY COMPOSITION

Assume that we are given $n \geq 1$ demonstration datasets $\mathcal{D}^{(1)}, \mathcal{D}^{(2)}, \dots, \mathcal{D}^{(n)}$, each $\mathcal{D}^{(l)}$ consisting of state–action pairs, $(\mathbf{s}, \mathbf{a}) \in \mathcal{D}^{(l)}$, without reward signal or temporal order. APC leverages these demonstrations to learn an overall policy $\pi(\mathbf{a} \mid \mathbf{s})$ by pre-training a set of prior-based actors, each based on its own dataset $\mathcal{D}^{(i)}$ (see Sec. 2.2), and then learning latent policies from reward feedback by interacting with the online RL task in *all* actors, including the prior-free actor. APC is a hierarchical reinforcement learning approach: The overall policy $\pi(\mathbf{a} \mid \mathbf{s})$ is a composition of the lower-level actors, controlled by a high-level selector that decides which actor to execute in each state. The prior-based actors solve tasks efficiently when demonstrations are aligned but can fail to achieve optimal performance under misalignment. In comparison, the prior-free actor lacks demonstration guidance but retains full flexibility to learn from reward feedback alone. This compositional architecture allows APC to both exploit distinct behavoir priors for efficient exploration and various aspects of the target task, while the prior-free actor can overcome limitations of the prior-based actors.

The remainder of this section is organized as follows: Sec. 3.1 formalizes the compositional policy model, consisting of multiple prior-based actors, the prior-free actor, and the high-level selector. An overview is provided in Fig. 1. Sec. 3.2 describes the online learning procedure for the lower-level actors, and Sec. 3.3 introduces a crucial technique for robust and efficient learning with multiple NF priors. Finally, Sec. 3.4 presents the key design of our high-level selector.

### 3.1 COMPOSITIONAL POLICY MODEL

Our policy model composes multiple lower-level actors to efficiently solve the online RL task. The higher-level selector $\pi_\beta$ is a conditional categorical distribution $\pi_\beta = \mathrm{Cat}(p_0(\mathbf{s}), p_1(\mathbf{s}), \dots, p_n(\mathbf{s}))$ with support $\beta \in \{0, 1, \dots, n\}$ that decides which lower-level actor to use in state $\mathbf{s}$. The values $p_i(\mathbf{s})$ are the probabilities of selecting the $i$-th lower-level actor in state $\mathbf{s}$, which means that $\pi_\beta(\beta = i \mid \mathbf{s}) = p_i(\mathbf{s})$. The set of lower-level actors consists of the prior-free actor $\pi_{\mathbf{a}}^{(0)}$ and $n$ prior-based actors $\pi_{\mathbf{a}}^{(l)}$, $1 \leq l \leq n$. The prior-free actor $\pi_{\mathbf{a}}^{(0)}$ uses an identity flow $T^{(0)}(\mathbf{z}; \mathbf{s}) = \mathbf{z}$ and the latent policy

$\pi_{\mathbf{z}}^{(0)} \colon \mathcal{S} \times \mathcal{Z} \to [0, 1]$, while each of the prior-based actors $\pi_{\mathbf{a}}^{(l)}$ consists of a NF $T^{(l)}$ that is pre-trained on the demonstration dataset $\mathcal{D}^{(l)}$ as described in Sec. 2.2 and a latent policy $\pi_{\mathbf{z}}^{(l)} \colon \mathcal{S} \times \mathcal{Z} \to [0, 1]$. To guarantee invertible flows, we also set $|\mathcal{Z}| = |\mathcal{A}|$. The overall policy's action distribution,

$$\pi(\mathbf{a} \mid \mathbf{s}) = \sum_{\beta'=0}^{n} \pi_\beta(\beta' \mid \mathbf{s}) \underbrace{\pi_{\mathbf{z}}^{(\beta')}\big(\tilde{T}^{(\beta'),}(\mathbf{a}; \mathbf{s}) \mid \mathbf{s}\big)}_{\text{likelihood of } \mathbf{a} \text{ under } \pi_{\mathbf{z}}^{(\beta')}} + \underbrace{\log |\det J_{\tilde{T}^{(\beta')}}(\mathbf{a}; \mathbf{s})|}_{\text{change of variables}}, \tag{3}$$

is a mixture over lower-level actors, weighted according to the selector $\pi_\beta$ and computed based on change of variables with the latent policies and pre-trained NFs. In equation 3 we use $\beta'$ to index latent policies and NFs for notational clarity.

APC uses MDP actions $\mathbf{a} \in \mathcal{A}$, selector actions $\beta \in \{0, 1, \ldots, n\}$ and latent actions $\mathbf{z} \in \mathcal{Z}$. Instead of sampling directly from the complex density $\pi(\mathbf{a} \mid \mathbf{s})$ in equation 3 at each time step $t$, we first obtain $\beta_t$ from the selector $\pi_\beta$ and then sample a latent action $\mathbf{z}_t$ from the chosen lower-level actor's latent policy $\pi_{\mathbf{z}}^{(\beta_t)}$, which we deterministically transform with the corresponding NF $T^{(\beta_t)}$ to obtain the action $\mathbf{a}_t$ for the online RL task. Our approach to learning $\pi(\mathbf{a} \mid \mathbf{s})$ is to learn the selector and each of the latent policies separately, as detailed in the following sections.

## 3.2 LOWER-LEVEL ACTOR LEARNING

As described above, our model contains $n + 1$ latent policies $\pi_{\mathbf{z}}^{(i)}$ that independently interact with the online RL task when they are chosen by the higher-level selector $\pi_\beta$ and thus observe transitions $(\mathbf{s}, \mathbf{z}, r, \mathbf{s}')$. We opt to run $n + 1$ parallel instances of Soft Actor-Critic (SAC) (Haarnoja et al., 2018), one for each of the lower-level actors, though any other off-policy RL algorithm that supports continuous action spaces could be used instead. We denote $\theta^{(i)}$ as the parameters for the latent policies and $\psi^{(i)}$ for the latent Q-functions. Each SAC optimizes an entropy-regularized RL objective and learns a unimodal Gaussian policy with so-called actor and critic updates. For more details on SAC, we refer to (Haarnoja et al., 2018). Note that the parameters $\phi^{(i)}$ of the pre-trained NFs $T^{(i)}$ are not updated during online learning, only the SAC parameters change, as per Singh et al. (2021). For more implementation and model details, we refer to App. C.

We maintain separate replay buffers $B^{(0)}, B^{(1)} \ldots, B^{(n)}$ for the $n + 1$ latent policies, which is motivated by the fact that the same latent coordinate $\mathbf{z}$ can correspond to different MDP actions $\mathbf{a}$ under the different NF behavior priors. This implies that the reward $r_t$ resulting from the latent action $\mathbf{z}_t$ only provides a valid feedback signal for the low-level actor that generated $\mathbf{a}_t$. Thus, at every time step $t$, we fill the replay buffer $B^{(\beta_t)}$ with the transition $(\mathbf{s}_t, \mathbf{z}_t, r_t, \mathbf{s}_{t+1})$ and update the parameters $\theta^{(\beta_t)}$ and $\psi^{(\beta_t)}$ of the *selected* lower-level actor on a batch of transitions sampled from buffer $B^{(\beta_t)}$. Next, we introduce a crucial mechanism that ensures both robust and efficient learning with multiple (prior-based) lower-level actors.

## 3.3 REWARD SHARING TRICK FOR BALANCED ACTOR UPDATES

The basic learning scheme outlined above updates only the latent policy $\pi_{\mathbf{z}}^{(\beta_t)}$ of the lower-level actor selected at step $t$. As the number of actors increases, this allocation of experience reduces sample efficiency: transitions are spread across multiple replay buffers, resulting in fewer updates per actor relative to the total number of environment interactions. More importantly, this setup can bias the higher-level selector: if a suboptimal prior-based actor initially outperforms, for example, the prior-free actor, the selector may overcommit to it, preventing crucial exploration of alternative actors that could potentially *later* outperform the actor that *initially* performs best.

To address this issue, we exploit the invertibility of NFs. Any executed action $\mathbf{a}_t$ can be mapped into the latent space of every prior-based actor via $\mathbf{z}_t^{(i)} = \tilde{T}^{(i)}(\mathbf{a}_t; \mathbf{s}_t)$, where the same environment action $\mathbf{a}_t$ gets mapped to *different* latent coordinates $\mathbf{z}_t^{(i)} \neq \mathbf{z}_t^{(j)}$. This happens because different demonstration datasets $\mathcal{D}^{(i)}$ and $\mathcal{D}^{(j)}$ induce different behavior priors and, as such, different transformations $\tilde{T}^{(i)}$ and $\tilde{T}^{(j)}$. Thus, using the inverse $\tilde{T}$ of these learned transformations, transitions can also be constructed for all actors $i \neq \beta_t$ that were not selected at time $t$, and their replay buffers $B^{(i)}$ can be populated with the constructed transitions. In this variant, each replay buffer receives a transition

at every step, enabling all actors to update continuously, independent of which one produced the executed action at time $t$. Our ablation results confirm that this feedback-sharing mechanism is essential: it not only improves sample efficiency but also prevents primacy bias – a tendency of RL algorithms to "*overfit early experiences that damages the rest of the learning process*" (Nikishin et al., 2022; Xu et al., 2024) – by ensuring fair learning progress for all actors.

### 3.4 HIGH-LEVEL SELECTOR

The high-level selector $\pi_\beta$ determines which lower-level actor to execute in each state. We adopt a learning-free *arbitrator* (Russell & Zimdars, 2003) design for implementing the selector, where the selection probabilities $p_l(\mathbf{s})$ are derived directly from the value estimates $V^{(l)}(\mathbf{s})$ of the lower-level actors. Concretely, the probability of selecting lower-level actor $l$ is

$$p_l(\mathbf{s}) = \frac{1}{Z}\exp\left(\frac{1}{\eta}V^{(l)}(\mathbf{s})\right), \quad Z = \sum_{i=0}^{n}\exp\left(\frac{1}{\eta}V^{(i)}(\mathbf{s})\right), \tag{4}$$

which defines a categorical distribution $\pi_\beta = \mathrm{Cat}(p_0(\mathbf{s}), p_1(\mathbf{s}), \ldots, p_n(\mathbf{s}))$, where $\eta$ is a temperature parameter that controls the sharpness of the distribution. However, because our lower-level agents are implemented with SAC, which does not provide a direct value function estimate, we approximate $V^{(l)}(\mathbf{s})$ via Monte Carlo estimates of each lower-level agent's $Q$-function. In practice, this amounts to sampling a single latent action $\mathbf{z}^{(l)}$ from each actor $l$ and evaluating $p_l(\mathbf{s}) = \frac{1}{Z}\exp(\frac{1}{\eta}Q^{(l)}(\mathbf{s}, \mathbf{z}^{(l)}))$, which yields an unbiased, though high-variance, estimate.

Implementing the selector as a learning-free arbiter provides two advantages: First, it eliminates the computational overhead of training a separate high-level neural network policy with additional parameters and gradient updates. Second, it circumvents the instabilities of hierarchical RL that arise when jointly optimizing higher- and lower-level agents, including sensitivity to learning-rate tuning, non-stationarity of the lower-level policies, primacy bias, and exploration difficulties. Our ablation results confirm that this design substantially improves stability and performance compared to learned higher-level policies.

## 4 EVALUATION AND ANALYSIS

We evaluate APC across continuous-control benchmarks to assess its robustness under demonstration misalignment and efficiency with access to demonstrations aligned with the online RL task. Our experiments address four central questions: **(i)** Can APC remain robust under severe demonstration misalignment, avoiding the performance degradation observed in prior methods? **(ii)** Can APC effectively exploit well-aligned demonstrations to accelerate learning? **(iii)** Can APC exceed the performance of suboptimal demonstrations? **(iv)** Which architectural components are critical for enabling such robust exploration?

### 4.1 ENVIRONMENTS

We test APC on the environments shown in Fig. 2. All environments, including data collection procedures and experimental setups, are described in greater detail in Appendix B.

**Maze Navigation:** Based on the well-known D4RL benchmark (Fu et al., 2020). Starting from the center, a point mass agent must reach different goal locations in a simple maze. We refer to the four goal locations as separate tasks. The state contains the agent's current position and velocity, and actions correspond to accelerations on the 2D plane.

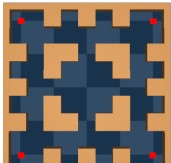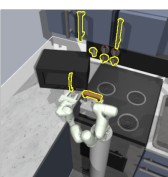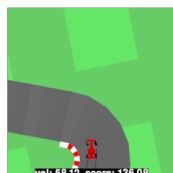

Figure 2: Our environmental testbed, from left to right: **Maze Navigation**, the different goals are marked in red. **Franka Kitchen**, with the manipulation targets marked in yellow. **Car Racing**.

**Franka Kitchen:** Based on (Gupta et al., 2019), a kitchen environment where a Franka Emika Panda robot needs to solve various manipulation tasks. The state is in $\mathbb{R}^{59}$ and contains symbolic

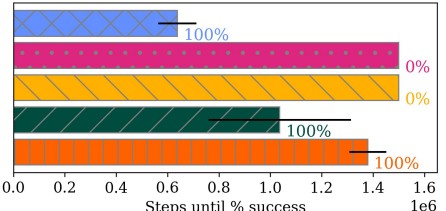
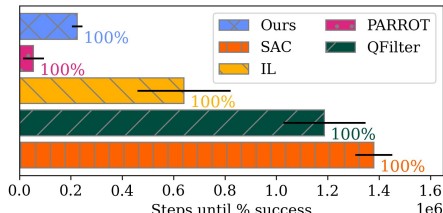

(a) Experiment **(i)**: Average performance on tasks with *misaligned* demonstrations, showing that PARROT and IL fail to learn, while APC solves the tasks even faster than from-scratch SAC, despite the misaligned prior.

(b) Experiment **(ii)**: Average performance on tasks with *aligned* demonstrations, showing that PAR-ROT solves the tasks most quickly, while APC considerably outperforms IL, QFilter, and from-scratch SAC.

Figure 3: Time to success in the PointMaze Navigation environment. Each method was executed for at most 1.5M environment steps; each experiment was repeated with three random seeds. Bars indicate the step at which the cross-seed average running success rate reached 100%, or the final success rate after 1.5M steps if convergence was not achieved earlier (shorter bars are better). Percentage annotations denote the cross-seed average running success rate at that time (3 seeds).

information about the agent and the manipulatable objects in the kitchen. Actions correspond to 9D joint velocities.

**Car Racing:** A top-down car racing environment from the Gymnasium suite (Towers et al., 2024). The task is to drive fast laps on the racing track, and actions correspond to steering and braking or accelerating.

## 4.2 BASELINES

We include the following baselines. **(SAC)** (Haarnoja et al., 2018) serves as a standard, from-scratch RL baseline for continuous action spaces. Comparing with this baseline reveals the acceleration in learning due to including (task-aligned) demonstration data and potential performance degradation due to misaligned demonstrations. Our next baseline is a simple yet powerful imitation learning **(IL)** approach Lu et al. (2023) that regularizes the policy by enforcing high likelihood for state-action pairs in the demonstration data. **(QFilter)** (Nair et al., 2018) is an extension of the IL baseline that only includes the imitation loss for $(\mathbf{s}, \mathbf{a})$ demonstration tuples that have a higher Q-value than the action sampled from the online learning policy in state $\mathbf{s}$. This baselines therefore also has the ability to exclude misaligned demonstrations from negatively affecting online performance. Lastly, **(PARROT)** (Singh et al., 2021) serves to reveal the increased adaptability of our method when using NF priors.

## 4.3 EXPERIMENTS & FINDINGS

**(i) APC shows robustness under demonstration misalignment:** We first evaluate robustness against demonstration misalignment in the PointMaze Navigation environment. Each method (except SAC) is provided with expert demonstrations $\mathcal{D}^{(i)}$ for one task from the four possible goals, and then evaluated on the remaining three tasks for which the demonstrations are misaligned. As shown in Fig. 3a, both PARROT and imitation learning (IL) fail to reliably solve the three tasks with the misaligned prior: after 1.5M steps their cross-seed running success rates remain at 0% and 7%, respectively. In contrast, APC reliably converges to 100% success in roughly 0.5M steps. Surprisingly, APC even outperforms from-scratch SAC despite the misaligned prior, indicating that APC can exploit misaligned priors for efficient exploration. Imitation learning combined with the QFilter is also able to avoid complete performance degradation due to the misaligned demonstrations on the PointMaze environment, but it learns considerably slower than APC, due to relying more heavily on accurate Q-function estimates and lacking the ability to exploit pre-trained behavior priors.

We observe consistent trends in the higher-dimensional FrankaKitchen environment. End-effector trajectories sampled from pre-trained NF priors (Fig. 4a) exhibit diverse behaviors, but similarity in trajectory geometry or task semantics is not predictive of transfer success (Fig. 4b). When

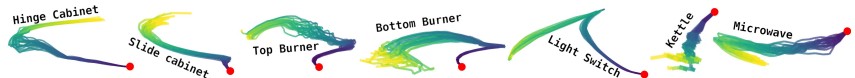

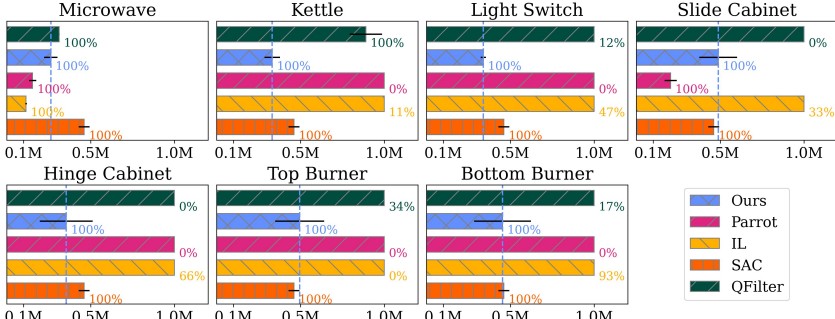

(a) 3D FrankaKitchen end-effector trajectories sampled from the per-task pre-trained NF priors. The red dot indicates the same starting state for each task; the trajectories have been shifted to save space.

(b) Time to success when using prior data from different tasks (panel titles) while optimizing the `microwave` task. Bars indicate the step at which the cross-seed average running success rate reached 100%, or the final success rate after 1M steps if convergence was not achieved earlier (shorter bars are better). Percentage annotations denote the cross-seed average running success rate at that time (3 seeds).

Figure 4: Results on FrankaKitchen's `microwave` task, which requires opening the microwave door. APC efficiently solves the task when exposed to aligned demonstrations (experiment **(ii)**, top-left panel) while remaining robust under demonstration misalignment (experiment **(i)**, remaining panels).

demonstrations are misaligned, PARROT and IL suffer severe losses in sample efficiency or fail entirely, underscoring their brittleness. APC, however, consistently solves the target task across all configurations, demonstrating strong robustness even in complex MDPs. This contrast highlights the importance of APC's ability to bypass misaligned priors, which is essential for adaptability under demonstration misalignment. Results for additional target tasks are reported in the appendix.

**(ii) APC efficiently exploits aligned priors:** We next consider the case where the demonstration prior is fully aligned with the online RL task. As expected, PARROT achieves the fastest convergence in this setting (Fig. 3, right; Fig. 4b, top left), since its behavior cloning pre-training step allows it to solve the task near-optimally just by random sampling in the latent space. APC, despite having access to the same near-optimal prior, must additionally learn accurate $Q$-function estimates for the arbiter to identify the beneficial prior, resulting in slightly slower convergence. Compared to the IL baseline, APC converges substantially faster on PointMaze and slightly slower on the FrankaKitchen environment, with neither method dominating overall. Importantly, all methods strictly outperform from-scratch SAC, confirming that all approaches are able to exploit the aligned demonstrations to accelerate learning. These results show that APC's increased adaptability under misalignment does not come at the cost of significant sample inefficiency in the aligned setting.

**(iii) APC exceeds suboptimal demonstrations:** We further evaluate APC in the CarRacing environment under a different but equally challenging form of demonstration misalignment. Instead of optimal demonstrations from related tasks, we collect $\approx 30$k transitions from a human driver on the target track. This dataset $\mathcal{D}$ is used to pre-train the NF behavior prior and is also provided to the IL baseline. The left panel in Fig. 5 shows the resulting return curves. SAC, trained from scratch, reaches optimal performance ($\approx 900$ return) after roughly 250k steps. PARROT, however, only marginally improves over the mean human score, indicating that the suboptimal prior imposes a strict performance ceiling. The IL baseline eventually surpasses human performance, but learning is slowed considerably by the imperfect demonstrations. QFilter also exceed the human performance and is less strongly affected than by the sub-optimal demonstrations than IL, achieving optimal returns after roughly 100k steps. APC achieves optimal returns in circa 30k steps, outperforming SAC and all demonstration-guided baselines. These results show that APC can exploit suboptimal priors to warm-start learning and guide exploration, while avoiding the performance ceilings that constrain existing

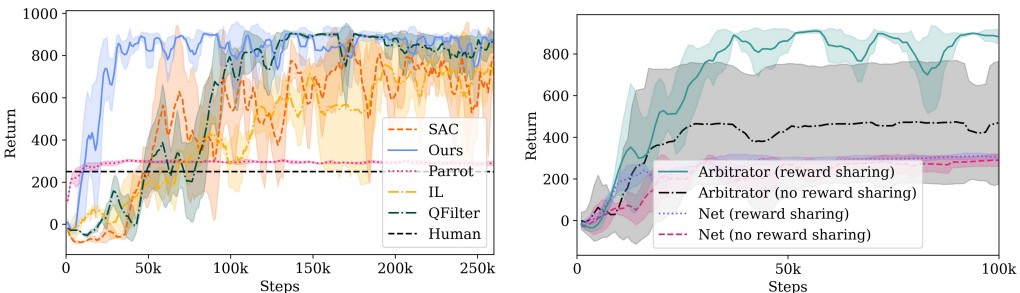

Figure 5: Return curves on the car racing environment, the shaded area corresponds to one standard deviation around the mean, averaged over three seeds. **Left**: Experiment **(iii)** showing APC's performance relative to our baselines. **Right**: Experiment **(iv)** shows the performance of multiple APC ablations.

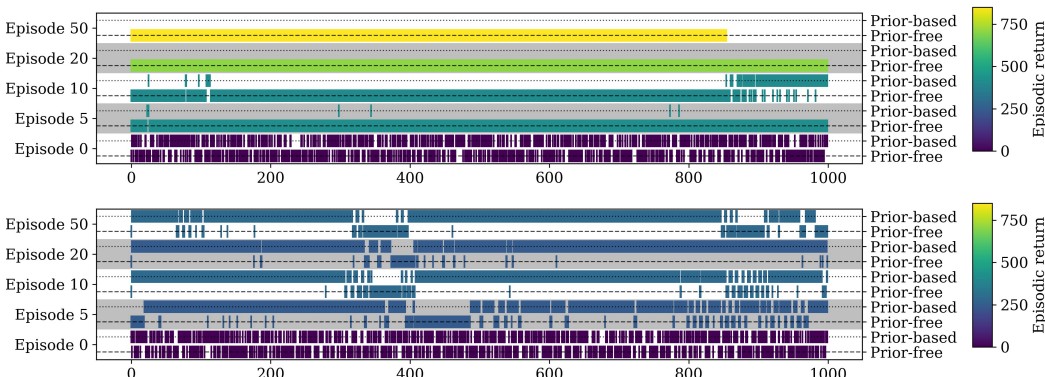

Figure 6: Qualitative visualization for experiment **(iv)** showing selector decisions on the CarRacing task. The rows in both panels correspond to selected evaluation episodes. Along each row, time steps progress from left to right, and each marker indicates whether the prior-based or prior-free actor was chosen at that step. **Top panel**: Decisions made by the parameter-free arbitrator selector with lower-level reward sharing. **Bottom panel**: Decisions made by a learned selector without reward sharing. This shows that the arbitrator selector exploits the prior-free actor to achieve high returns, while the learned selector overcommits to the prior-based actor and achieves subpar returns. An extended figure showing all ablations and more episodes can be found in App. E.

methods. This makes APC especially valuable in practical settings where suboptimal demonstrations are readily available, while expert demonstrations might be scarce.

**(iv) Arbitrator and reward sharing are crucial for exploration:** To disentangle the contributions of the arbitrator selector architecture (Sec. 3.4) and the reward-sharing trick for efficient learning (Sec. 3.3), we performed an ablation study in the CarRacing environment.

We compare four variants: (i) our full method with an arbitrator selector and reward sharing, (ii) an arbitrator selector without reward sharing, (iii) a hierarchical *learned* selector with reward sharing between the latent actors, and (iv) a hierarchical *learned* selector without reward sharing between the latent actors. The learned hierarchical selector is also optimized with SAC and attempts to maximize the same environment reward as the latent actors. Its action space is discrete, parameterizing a Categorical distribution representing the choice over which lower-level actor to execute at each step $t$.

The corresponding return curves in the right panel of Fig. 5 (right) show a clear separation: only our full method consistently achieves optimal return. Two observations explain these results. First, replacing the arbitrator-style selector with a learned high-level policy introduces strong primacy bias. As shown in Fig. 6 (bottom), the learned selector overcommits to using the prior-based actor, as it initially achieves higher return than the randomly initialized prior-free actor. Once this bias is reinforced, the prior-free actor is rarely used and cannot quickly improve, even though it could

ultimately surpass the NF prior. In contrast, the arbitrator avoids this failure mode by directly comparing value estimates across actors, without learning an additional policy.

However, second, the arbitrator alone does not suffice. Without reward sharing, transitions collected by the prior-based actor benefit only that actor, further amplifying its dominance. Using the arbitrator alongside reward sharing (Fig. 6, top) ensures that all actors are updated on every transition, enabling the prior-free actor to learn from higher-quality trajectories produced by the prior-based actor. This allows it to rapidly improve its $Q$-value estimates and eventually outperform the prior-based actor.

These results show that both mechanisms are critical: the arbitrator mitigates primacy bias, while reward sharing ensures fair competition through data-efficient learning. Our design combines both to achieve robust exploration and accelerated learning without premature convergence to suboptimal actors.

## 4.4 SUMMARY OF RESULTS

Across environments and settings, our experiments demonstrate three consistent findings. First, APC remains robust under misaligned demonstrations, reliably solving target tasks where PARROT and IL fail, and in some cases even outperforming from-scratch SAC by exploiting misaligned priors for exploration. Second, APC effectively leverages aligned demonstrations: while slightly slower than PARROT under perfectly aligned priors, APC consistently outperforms from-scratch SAC, showing that its added flexibility does not compromise sample efficiency. Third, APC exceeds suboptimal demonstrations by bootstrapping from imperfect data without imposing performance ceilings from the data. Finally, our ablation studies confirm that the arbitrator-style selector and reward-sharing mechanism are both necessary to prevent primacy bias and ensure fair, data-efficient competition among actors. Together, these results highlight APC's robustness, efficiency, and adaptability across diverse demonstration settings.

## 5 RELATED WORK

**Normalizing Flows for RL** In addition to using pre-trained NFs as data-driven action priors, NFs have also been used to replace simple unimodal Gaussian policies with richer, multimodal distributions, with the reported benefit of improved exploration and sample efficiency (Ward et al., 2019; Mazoure et al., 2020). In these approaches, the NF parameters are learned jointly with the policy during online RL, effectively treating the flow as part of the policy network. In contrast, our method keeps the NF prior fixed after pre-training it on demonstrations. NFs have also been leveraged to enforce safety constraints by mapping the action space into a constraint-respecting action subspace (Brahmanage et al., 2023; Chen et al., 2023), similar to "invalid action masking" techniques (Kalweit et al., 2020; Huang & Ontañón, 2022; Rietz et al., 2024). Our use of flows differs from these works: rather than masking out forbidden actions, we bias the policy by searching in the NF prior's latent space to guide exploration towards behaviors observed in the demonstration data. Using pre-trained NFs as demonstration-driven action priors was introduced by Singh et al. (2021), who argue that the invertible nature of NFs allows for flexible adaptation to the online task. Our results, however, show that a misaligned NF prior is hard to escape in practice, and that our hierarchical design, which explicitly allows the agent to bypass misaligned priors, greatly increases adaptability and robustness under distribution shift.

**Skill-based Hierarchical Learning** A large body of work exploits hierarchical architectures to accelerate learning by introducing temporal abstraction. In these approaches, a high-level policy selects between discrete options or "primitives" (Sutton et al., 1999; Fox et al., 2017; Ajay et al., 2020; Kulkarni et al., 2016), which may be obtained from demonstrations, learned through unsupervised exploration (Eysenbach et al., 2018; Park et al., 2024; 2023), or provided as scripted controllers (Nasiriany et al., 2022; Chitnis et al., 2020; Sharma et al., 2020). Other methods instead construct a continuous latent embedding of skills" (Pertsch et al., 2021; Yang et al., 2022; Rana et al., 2022), and solve downstream tasks by searching over the latent space. The works lack means for adapting when the primitives or skills do not suffice for solving the task. An exception to this is MAPLE (Nasiriany et al., 2022), which also learns an online policy to improve upon the given scripted controllers. Our work differs from these since we do not leverage temporal abstraction for exploration but instead focus on robustness and adaptability under demonstration misalignment.

**Offline to Online RL**   A complementary line of work accelerates online RL by leveraging pre-collected offline datasets of MDP transitions $\mathcal{D} = \langle \mathbf{s}, \mathbf{a}, r, \mathbf{s}' \rangle_{i=1}^{N}$ (Levine et al., 2020; Xie et al., 2021). Ball et al. (2023) balance online and offline data through joint sampling in off-policy RL. Nair et al. (2020) pre-train a policy offline and constrain the subsequent online policy to remain close to it. Zhang et al. (2023); Hu et al. (2024) pre-train both the policy and value function, and then refine the value function online while using both the offline and online policies as proposal distributions. Kong et al. (2024) adopt a similar proposal-policy scheme but periodically reset the online policy to counteract primacy bias. While effective, these methods primarily target the distributional shift between offline and online RL. Crucially, they also require reward-labeled data for pre-training, whereas our approach relies only on unlabeled demonstrations to train NF priors.

**Learning from Demonstrations**   Learning from demonstration has a long history in RL. Most approaches incorporate demonstrations through explicit imitation losses that encourage the policy to stay close to the demonstrated behavior (Ross et al., 2011; Hester et al., 2018; Goecks et al., 2019; Fujimoto & Gu, 2021; Lu et al., 2023; Tiapkin et al., 2024), generally assuming that demonstrations are aligned with the target task and offer no mechanism to cope with substantially misaligned demonstrations. Inverse RL methods infer the reward function from demonstrations and subsequently optimize it with RL (Abbeel & Ng, 2004; Ziebart et al., 2008; Ho & Ermon, 2016), but likewise depend on demonstrations that are near-optimal. Some recent works can account for suboptimal demonstrations (Nair et al., 2018; Zhao et al., 2022; Hu et al., 2024; Dong et al., 2025; Cramer et al., 2025), but lack APC's ability to adaptive compose multiple distinct behavior priors with a prior-free actor.

## 6   LIMITATIONS AND DISCUSSIONS

An apparent shortcoming of APC lies in its high computational overhead that scales linearly with the number of latent actors, since each actor is updated separately with SAC. While reward sharing improves sample efficiency, maintaining multiple parallel learners increases wall-clock time and limits scalability to larger sets of behavior priors.

Although APC is designed to remain robust under demonstration misalignment and distribution shift, it may still fail in adversarial or contrived scenarios. If many severely misaligned priors all bias exploration toward task-*irrelevant* regions of the state-space, and if the reward signal is uninformative about this sub-optimality, then each actor's Q-values might not allow the arbitrator to distinguish and avoid misaligned behaviors.

## 7   FUTURE WORK

We see addressing the computational demands of APC as important future work. This could be approached by maintaining and updating a shared, central critic, while heuristically updating only the *selected* actor at time $t$, instead of updating all full actor-critics at each step. This might substantially reduce computational overhead and wall-clock time, while preserving APC's adaptive behavior and robustness under demonstration misalignment and distribution shift.

## 8   CONCLUSION

This paper proposes Adaptive Policy Composition (APC), a hierarchical RL architecture that composes multiple NF priors with a prior-free fallback actor under an adaptive selector. By combining a parameter-free arbitrator with reward sharing, APC ensures data-efficient learning across all actors and avoids primacy bias, enabling robust demonstration-guided exploration even under misalignment. Our experiments across diverse benchmarks show that APC leverages aligned demonstrations, remains robust under misalignment, and exceeds suboptimal demonstrations by using priors to bootstrap exploration. These findings demonstrate that APC is a general approach for integrating imperfect demonstrations into online RL *without* impairing performance, thereby bridging the gap between data-driven priors and reward-driven adaptation.

ACKNOWLEDGMENTS

This work was partially supported by the Wallenberg AI, Autonomous Systems and Software Program (WASP) funded by the Knut and Alice Wallenberg Foundation.

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

## A  REPRODUCIBILITY STATEMENT

Source code is available at https://github.com/frietz58/apc-rl.

## B  ENVIRONMENT DETAILS

### B.1  MAZE NAVIGATION

We adopt the maze navigation environment from D4RL (Fu et al., 2020); however, we customize the maze layout as shown in Fig. 2. The agent corresponds to a simple point mass, with actions $\mathcal{A} \in \mathbb{R}^2$ corresponding to linear force exerted on the point. The observation space $\mathcal{S} \in \mathbb{R}^4$ contains the agent's current $(x, y)$ position and velocity. The task encoding, defined by one of four distinct goal locations, is not part of the observation and must be inferred from the reward signal. This still yields a standard, fully observable MDP for each separate task.

The reward function is dense and defined as the exponential of the negative Euclidean distance between the agent and the goal. To encourage short episodes, we subtract a constant penalty of $-1$ at each step. Episodes start from a random position near the maze center, terminate successfully when the agent reaches within 0.5 units of the goal, and are truncated after 400 steps.

For each goal location $i$, we generate demonstration datasets $\mathcal{D}^{(1)}, \ldots, \mathcal{D}^{(4)}$ using extensively pre-trained, optimal policies $\pi^{(i)*}$. Specifically, we collect 100 episodes per task by sampling actions from $\pi^{(i)*}$ and recording the resulting $(\mathbf{s}, \mathbf{a})$ pairs. These datasets are used either to pre-train the NF priors (one per task) or directly as input to the IL baseline, depending on the evaluation setting.

### B.2  FRANKAKITCHEN

We use the FrankaKitchen environment introduced by Gupta et al. (2019), which features seven distinct manipulation tasks: opening the `microwave` door, pushing the `kettle` onto the correct stove burner, turning on the `bottom burner` by rotating the corresponding knob, turning on the `top burner` by rotating the corresponding knob, flipping the `light switch` to the on position, opening the `sliding cabinet` door, and opening the `hinge cabinet` door. The state space $\mathcal{S} \in \mathbb{R}^{59}$ contains symbolic features describing all manipulable objects, along with the robot's joint angles and velocities. The action space $\mathcal{A} \in \mathbb{R}^9$ corresponds to joint velocity commands. The one-hot task identity is not included in the state and must instead be inferred from the reward signal, yielding a standard, fully observable MDP for each individual task.

To facilitate exploration and accelerate training, we replace the original sparse rewards with a dense reward function. This modification was necessary given the high computational burden of evaluating a combinatorial number of tasks and prior settings for multiple seeds. Let $\mathbf{p}_{\text{ee}} \in \mathbb{R}^3$ denote the end-effector position, computed as the midpoint of the left and right gripper fingers, and let $\mathbf{p}_{\text{obj}} \in \mathbb{R}^3$ denote the position of the target object for the current task. We define the end-effector distance term as

$$r_{\text{ee}} = -\alpha, \|\mathbf{p}_{\text{ee}} - \mathbf{p}_{\text{obj}}\|_2, \tag{5}$$

with scaling factor $\alpha = 0.5$. For each task $k$, the environment additionally provides an achieved goal state $\mathbf{g}_{\text{ach}}^{(k)}$ and a desired goal state $\mathbf{g}_{\text{des}}^{(k)}$. We can thus compute a task success distance term as

$$r_{\text{task}} = -\|\mathbf{g}_{\text{ach}}^{(k)} - \mathbf{g}_{\text{des}}^{(k)}\|_2, \tag{6}$$

which encourages the agent to bring the target object into its goal configuration (e.g., microwave door fully open). Our final dense reward function is then given by

$$r(\mathbf{s}, \mathbf{a}) = \begin{cases} R_{\text{success}}, & \text{if } |r_{\text{task}}| \leq \epsilon, \\ r_{\text{ee}} + r_{\text{task}}, & \text{otherwise,} \end{cases} \tag{7}$$

where $R_{\text{success}} = 100$ is a large completion bonus.

For each of the seven tasks $i$, we construct demonstration datasets $\mathcal{D}^{(1)}, \ldots, \mathcal{D}^{(7)}$ using extensively pre-trained, optimal policies $\pi^{(i)*}$. Each dataset consists of 100 episodes collected by executing $\pi^{(i)*}$ and recording the resulting $(\mathbf{s}, \mathbf{a})$ pairs. These datasets are either used to pre-train task-specific NF priors or passed directly to the IL baseline, depending on the evaluation setting.

### B.3 CARRACING

We use the CarRacing environment from the Gymnasium suite (Towers et al., 2024), which requires driving fast laps on a top-down race track. The simulated planar car follows simplified vehicle dynamics that include skidding and varying friction across terrain types (asphalt vs. grass). The continuous action space is $\mathcal{A} \in \mathbb{R}^3$, corresponding to steering, acceleration, and braking. While the original environment provides pixel observations, we extract a symbolic representation directly from the simulator.

The symbolic observation space captures the vehicle's relative position, orientation, and motion with respect to the track. At each timestep, the agent is exposed to the following symbols:

- Track-edge distances: signed distances to the left and right road boundaries, $(d_{\text{left}}, d_{\text{right}})$.
- Heading error: orientation difference $\Delta\theta$ between the car's heading and the tangent of the nearest track centerline, wrapped to $[-\pi, \pi]$.
- Velocities: forward and lateral velocity components in the car's local frame, $(v_{\text{fwd}}, v_{\text{side}})$, and the angular velocity $\omega$.
- Lookahead waypoints: relative positions of the next $L = 5$ centerline waypoints in the car's local coordinate frame, $\{(x_j, y_j)\}_{j=1}^{L}$.

Formally, the observation vector is

$$\mathbf{s} = \begin{bmatrix} d_{\text{left}}, \ d_{\text{right}}, \ \Delta\theta, \ v_{\text{fwd}}, \ v_{\text{side}}, \ \omega, \ x_1, y_1, \ldots, x_L, y_L \end{bmatrix} \in \mathbb{R}^{6+2L}, \qquad (8)$$

which yields $\mathcal{S} \in \mathbb{R}^{16}$ for $L = 5$. Episodes begin with the car at rest at a fixed position centered on the track, and we enforce deterministic resets such that the track layout remains identical across episodes.

We use the environment's unmodified reward function: each step incurs a penalty of $-0.1$, and the agent receives a reward of $+1000/M$, where $M$ is the number of track-tiles visited during the current episode.

This environment contains only a single task – driving efficiently on the fixed track. We collect a demonstration dataset $\mathcal{D}^{(1)}$ by recording 10 trajectories from a human driver, with an average return of approximately 250. This dataset is used either to pre-train the NF prior or directly as input to the IL baseline.

## C  TRAINING AND MODEL DETAILS

### C.1  SAC

Our main learning algorithm is Soft Actor-Critic (SAC) (Haarnoja et al., 2018). We follow the standard learning procedure described by Haarnoja et al. (2018) without modification, and use largely the same hyperparameters across environments (Tab. 1), with minor adjustments to the discount factor and Polyak target coefficient to stabilize training, particularly in the CarRacing environment with its high-magnitude rewards. SAC is employed in all of our experiments: (i) as a from-scratch baseline, (ii) to implement the IL baseline, and (iii) to train the latent policies of the lower-level actors within APC and PARROT. To make for a fair and consistent comparison, the same SAC hyperparameters are used for all methods in each experiment.

### C.2  NORMALIZING FLOW BEHAVIOR PRIOR

We implement the NF prior using a conditional version of the real NVP architecture (Dinh et al., 2017), which is composed of multiple affine coupling layers. Each affine coupling layer splits the input $\mathbf{x} \in \mathbb{R}^D$ into two parts and computes the output $\mathbf{y}$ by applying a scale-and-shift transformation to one part, conditioned on the other:

$$\mathbf{y}_{[1:d]} = \mathbf{x}_{[1:d]}, \qquad \mathbf{y}_{[d+1:D]} = \mathbf{x}_{[d+1:D]} \odot \exp\left(v(\mathbf{x}_{[1:d]}, \mathbf{s})\right) + q(\mathbf{x}_{[1:d]}, \mathbf{s}), \qquad (9)$$

where $v$ and $q$ are neural networks that additionally take the state $\mathbf{s}$ as input, to learn different transformations in different states. Concretely, we implement $v$ and $q$ as fully connected MLPs, with

Table 1: SAC hyperparameters used across environments.

| Hyperparameter | PointMaze | FrankaKitchen | CarRacing |
|---|---|---|---|
| Number of parallel environments | 5 | 10 | 1 |
| Replay buffer size | $1 \times 10^6$ | $1 \times 10^6$ | $1 \times 10^6$ |
| Discount factor $\gamma$ | 0.995 | 0.995 | 0.995 |
| Polyak target coefficient $\tau$ | 0.005 | 0.01 | 0.005 |
| Batch size | 256 | 256 | 256 |
| Learning starts | $1 \times 10^3$ | $1 \times 10^3$ | $1 \times 10^3$ |
| Policy learning rate | $3 \times 10^{-4}$ | $3 \times 10^{-4}$ | $3 \times 10^{-4}$ |
| Q-function learning rate | $1 \times 10^{-3}$ | $1 \times 10^{-3}$ | $1 \times 10^{-3}$ |
| Entropy coefficient $\alpha$ | 0.1 | 0.1 | 0.005 |
| Entropy autotune | `False` | `False` | `False` |
| Actor network type | Fully-connected | Fully-connected | Fully-connected |
| Actor hidden layer widths | [512, 256] | [512, 256] | [512, 256] |
| Actor optimizer | Adam | Adam | Adam |
| Actor activation function | `tanh` | `tanh` | `tanh` |
| Critic network type | Fully-connected | Fully-connected | Fully-connected |
| Critic hidden layer widths | [512, 256] | [512, 256] | [512, 256] |
| Critic optimizer | Adam | Adam | Adam |
| Critic activation function | `tanh` | `tanh` | `tanh` |

hyperparameters summarized in Tab. 2. To increase expressivity, we interleave each affine coupling layer with a parameter-free flip transformation layer that reverses the order of the input dimensions.

**Pre-training.** Each NF prior is pre-trained on a demonstration dataset $\mathcal{D}$ using maximum-likelihood estimation (Eq. 2). To improve stability, particularly in settings with low-variance or near-unimodal action distributions, we add two regularization terms: (i) an inverse-consistency penalty $\mathcal{L}_{\text{ic}}$ encouraging nearby actions in real space to map to similar latent codes, and (ii) a forward-smoothness penalty $\mathcal{L}_{\text{fs}}$ encouraging local smoothness in the mapping from latent to real actions. The overall loss for training the NF prior is

$$\mathcal{L}(\phi) = -\mathbb{E}_{(\mathbf{a},\mathbf{s})\sim\mathcal{D}}\Big[\log\mathcal{N}\big(\tilde{T}_\phi(\mathbf{a};\mathbf{s});\mathbf{0},\mathbf{I}\big) + \log\big|\det J_{\tilde{T}_\phi}(\mathbf{a};\mathbf{s})\big| + \lambda_{\text{ic}}\,\mathcal{L}_{\text{ic}} + \lambda_{\text{fs}}\,\mathcal{L}_{\text{fs}},\Big] \quad (10)$$

where

$$\mathcal{L}_{\text{ic}} = \mathbb{E}_{\mathbf{a},\mathbf{s},\boldsymbol{\epsilon}_a}\left[\frac{\|\tilde{T}_\phi(\mathbf{a}+\boldsymbol{\epsilon}_a;\mathbf{s}) - \tilde{T}_\phi(\mathbf{a};\mathbf{s})\|_2^2}{\|\boldsymbol{\epsilon}_a\|_2^2 + \varepsilon}\right], \quad (11)$$

$$\mathcal{L}_{\text{fs}} = \mathbb{E}_{\mathbf{a},\mathbf{s},\boldsymbol{\delta}_z}\left[\frac{\|T_\phi(\mathbf{z}+\boldsymbol{\delta}_z;\mathbf{s}) - T_\phi(\mathbf{z};\mathbf{s})\|_2^2}{\|\boldsymbol{\delta}_z\|_2^2 + \varepsilon}\right], \quad (12)$$

where $\boldsymbol{\epsilon}_a$ and $\boldsymbol{\delta}_z$ are noise vectors sampled from zero-centered Gaussians with standard deviation 0.01, and $\varepsilon$ is a small term for numerical stability. $\lambda_{\text{ic}}$ and $\lambda_{\text{fs}}$ control the strength of the respective penalties.

**Online usage.** During the online RL phase, the NF priors are used only for inference: a latent action $\mathbf{z}_t$ sampled from a latent policy is transformed into an environment action $\mathbf{a}_t = T(\mathbf{z}_t;\mathbf{s}_t)$. Optionally, via the feedback-sharing mechanism (Sec. 3.3), the inverse mapping $\tilde{T}$ is applied to compute latent codes for other actors' policies. Importantly, the latent policies never backpropagate through the NF prior. From their perspective, the NF is simply part of the environment and affects the transition and reward dynamics.

For additional background on real NVPs and their use as behavior priors in RL, we refer to Dinh et al. (2017); Singh et al. (2021).

Table 2: Normalizing Flow (NF) prior hyperparameters used across environments.

| Hyperparameter | PointMaze | FrankaKitchen | CarRacing |
|---|---|---|---|
| Number of coupling layers | 10 | 10 | 10 |
| Hidden layer widths of $q$, $v$ | [256] | [256] | [256] |
| Activation function | ReLU | ReLU | ReLU |
| Base distribution covariance | 0.2 | 0.2 | 0.2 |
| Learning rate | $1 \times 10^{-4}$ | $1 \times 10^{-4}$ | $1 \times 10^{-4}$ |
| Batch size | 64 | 64 | 1024 |
| Number of training epochs | 100 | 100 | 100 |
| Gradient clipping norm | 1.0 | 1.0 | 1.0 |
| Inverse-consistency penalty $\lambda_{\text{ic}}$ | $1 \times 10^{-3}$ | $1 \times 10^{-3}$ | $1 \times 10^{-3}$ |
| Forward-smoothness penalty $\lambda_{\text{fs}}$ | $1 \times 10^{-3}$ | $1 \times 10^{-3}$ | $1 \times 10^{-3}$ |
| Optimizer | Adam | Adam | Adam |

## D ADDITIONAL FRANKAKITCHEN RESULTS

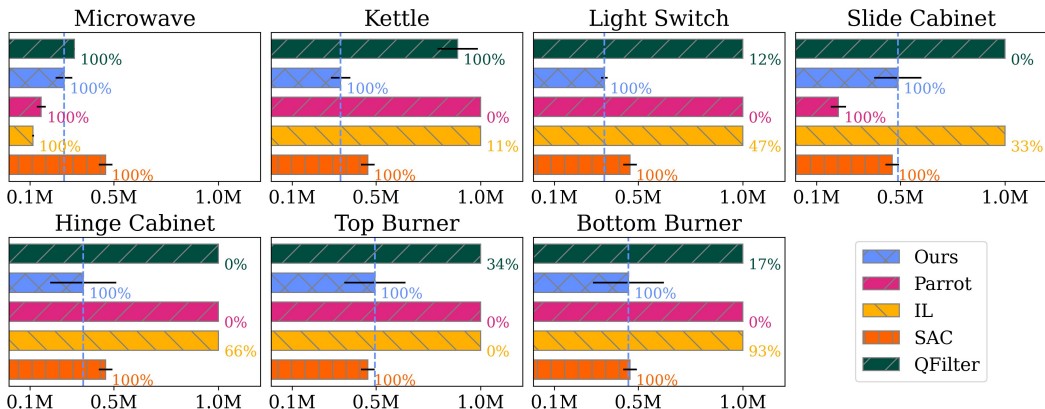

(a) Replica of Fig. 4b. Time to success when using prior data $\mathcal{D}_j$ from different tasks (panel titles). Bars indicate the step at which the cross-seed average running success rate reached 100%, or the final success rate after 1M steps if convergence was not achieved earlier (shorter is better). Percentage annotations denote the cross-seed average running success rate at that time (3 seeds). Dashed vertical lines indicate the convergence time of APC.

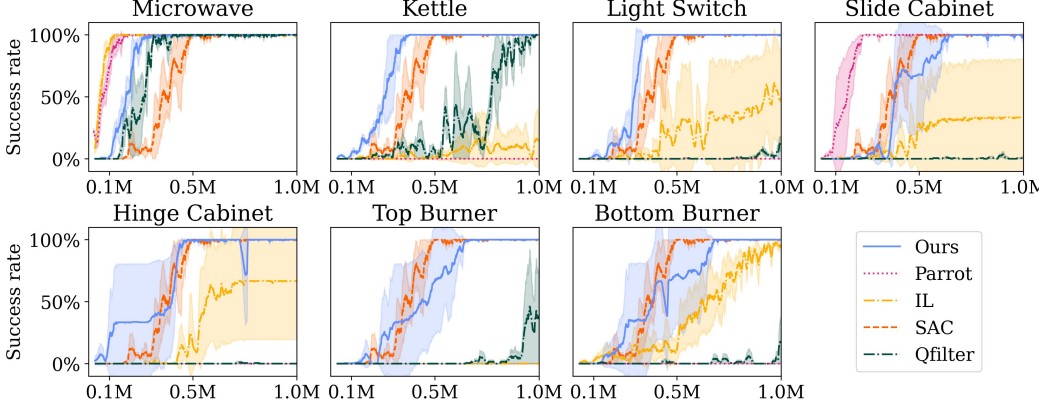

(b) Success rate over time corresponding to the above bar plot, using prior data from different tasks (panel titles). The shaded area corresponds to one standard deviation across three random seeds.

Figure 7: Results on FrankaKitchen's `microwave` task, which requires opening the microwave door.

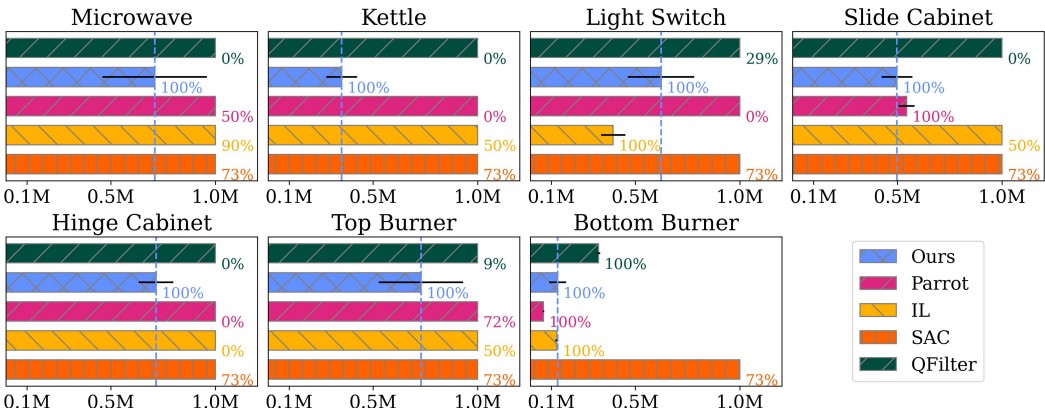

(a) Time to success when using prior data $\mathcal{D}_j$ from different tasks (panel titles). Bars indicate the step at which the cross-seed average running success rate reached 100%, or the final success rate after 1M steps if convergence was not achieved earlier (shorter is better). Percentage annotations denote the cross-seed average running success rate at that time (2 seeds). Dashed vertical lines indicate the convergence time of APC.

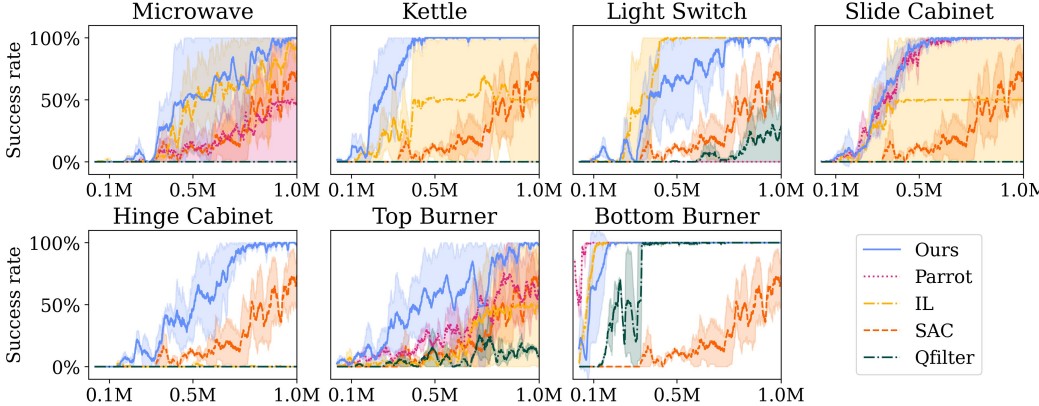

(b) Success rate over time corresponding to the above bar plot, using prior data from different tasks (panel titles). The shaded area corresponds to one standard deviation across two random seeds.

Figure 8: Results on FrankaKitchen's `bottom burner` task, which requires turning the knob to turn on one of the bottom-row stove burners.

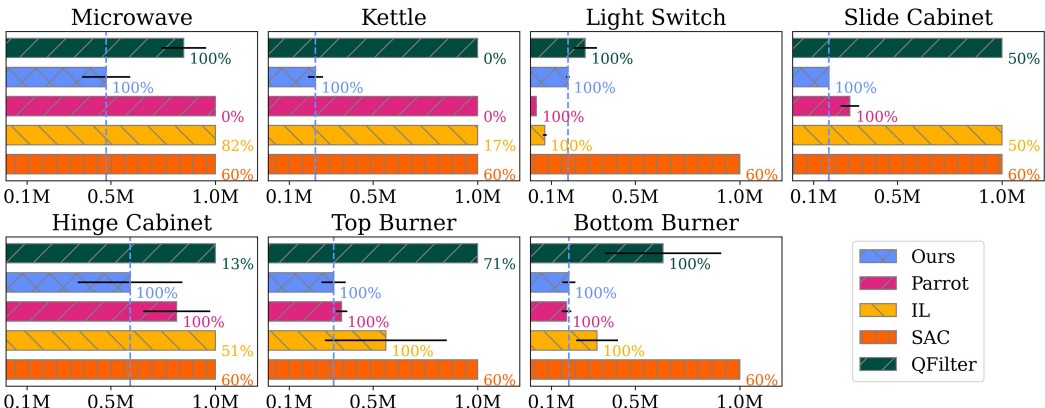

(a) Time to success when using prior data $\mathcal{D}_j$ from different tasks (panel titles). Bars indicate the step at which the cross-seed average running success rate reached 100%, or the final success rate after 1M steps if convergence was not achieved earlier (shorter is better). Percentage annotations denote the cross-seed average running success rate at that time (2 seeds). Dashed vertical lines indicate the convergence time of APC.

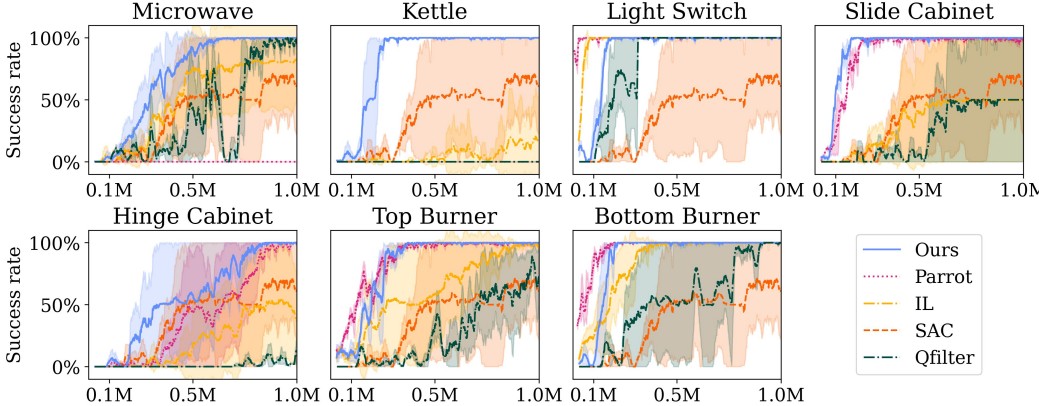

(b) Success rate over time corresponding to the above bar plot, using prior data $\mathcal{D}_j$ from different tasks (panel titles). The shaded area corresponds to one standard deviation across two random seeds.

Figure 9: Results on FrankaKitchen's `light switch` task, which requires flipping the light switch up.

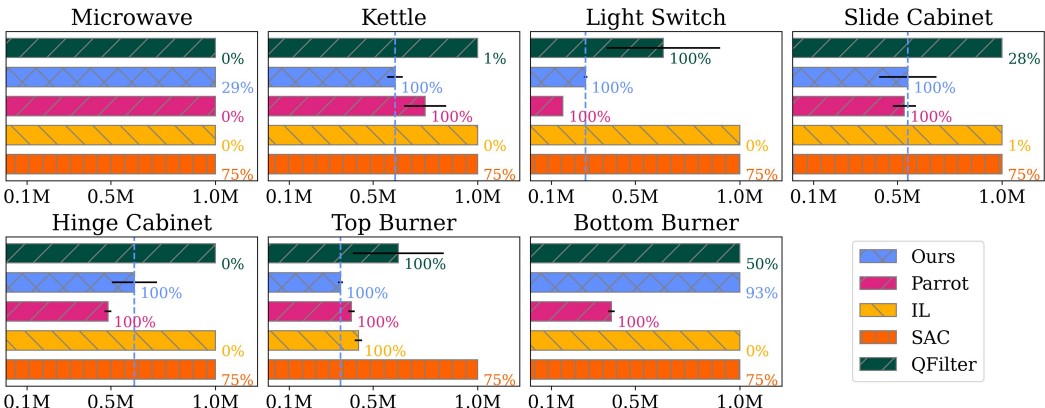

(a) Time to success when using prior data $\mathcal{D}_j$ from different tasks (panel titles). Bars indicate the step at which the cross-seed average running success rate reached 100%, or the final success rate after 1M steps if convergence was not achieved earlier (shorter is better). Percentage annotations denote the cross-seed average running success rate at that time (2 seeds). Dashed vertical lines indicate the convergence time of APC.

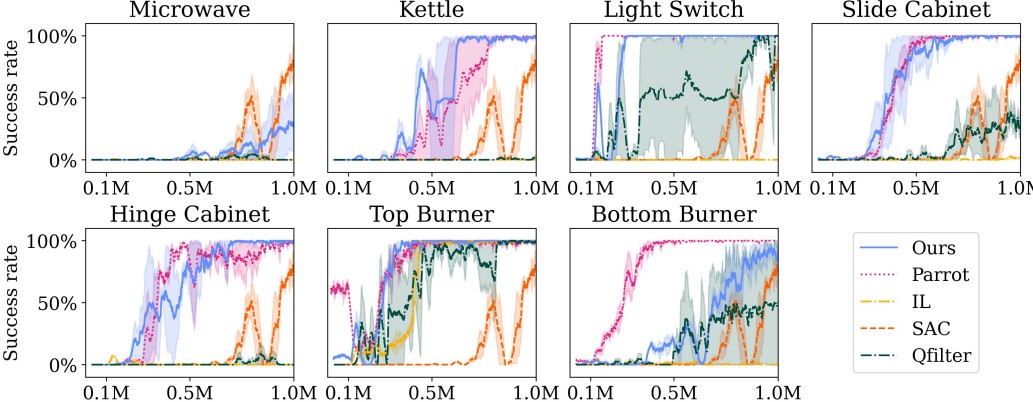

(b) Success rate over time corresponding to the above bar plot, using prior data $\mathcal{D}_j$ from different tasks (panel titles). The shaded area corresponds to one standard deviation across two random seeds.

Figure 10: Results on FrankaKitchen's `top burner` task, which requires turning the know to turn on one of the top-row stove burners.

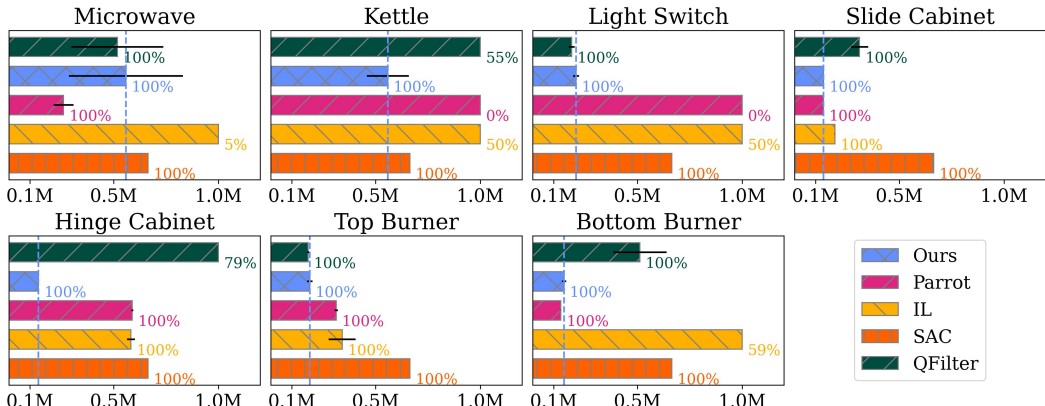

(a) Time to success when using prior data $\mathcal{D}_j$ from different tasks (panel titles). Bars indicate the step at which the cross-seed average running success rate reached 100%, or the final success rate after 1M steps if convergence was not achieved earlier (shorter is better). Percentage annotations denote the cross-seed average running success rate at that time (2 seeds). Dashed vertical lines indicate the convergence time of APC.

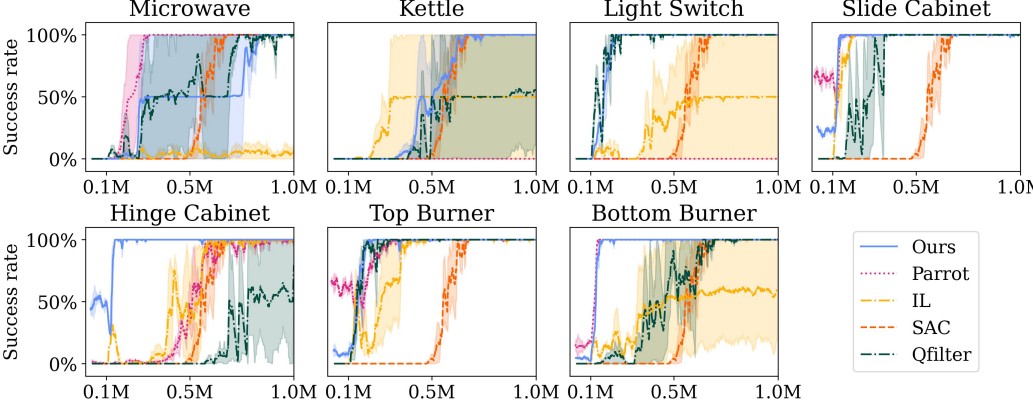

(b) Success rate over time corresponding to the above bar plot, using prior data $\mathcal{D}_j$ from different tasks (panel titles). The shaded area corresponds to one standard deviation across two random seeds.

Figure 11: Results on FrankaKitchen's `slide cabinet` task, which requires sliding open the top-right cabinet door.

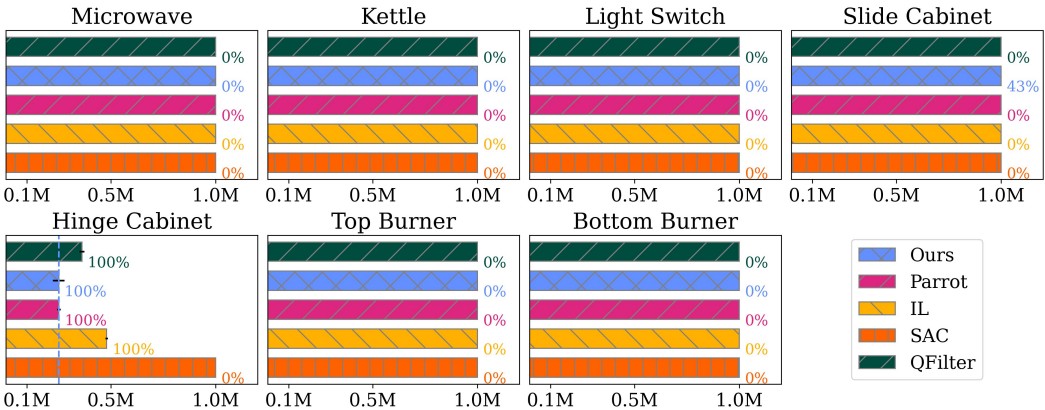

(a) Time to success when using prior data $\mathcal{D}_j$ from different tasks (panel titles). Bars indicate the step at which the cross-seed average running success rate reached 100%, or the final success rate after 1M steps if convergence was not achieved earlier (shorter is better). Percentage annotations denote the cross-seed average running success rate at that time (2 seeds). Dashed vertical lines indicate the convergence time of APC.

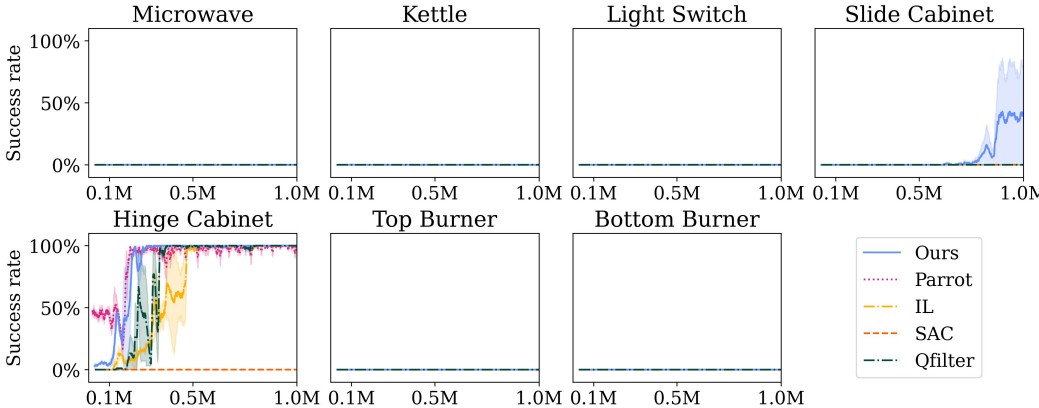

(b) Success rate over time corresponding to the above bar plot, using prior data $\mathcal{D}_j$ from different tasks (panel titles). The shaded area corresponds to one standard deviation across two random seeds.

Figure 12: Results on FrankaKitchen's `hinge cabinet` task, which requires opening the top-left "hinge" type cabinet door. Due to the difficulty of the task, all methods fails at solving the task when not using an aligned prior.

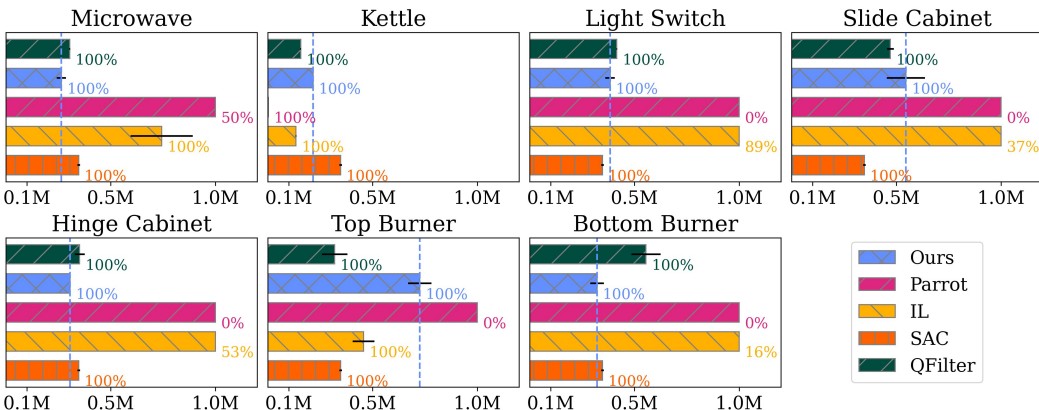

(a) Time to success when using prior data $\mathcal{D}_j$ from different tasks (panel titles). Bars indicate the step at which the cross-seed average running success rate reached 100%, or the final success rate after 1M steps if convergence was not achieved earlier (shorter is better). Percentage annotations denote the cross-seed average running success rate at that time (2 seeds). Dashed vertical lines indicate the convergence time of APC.

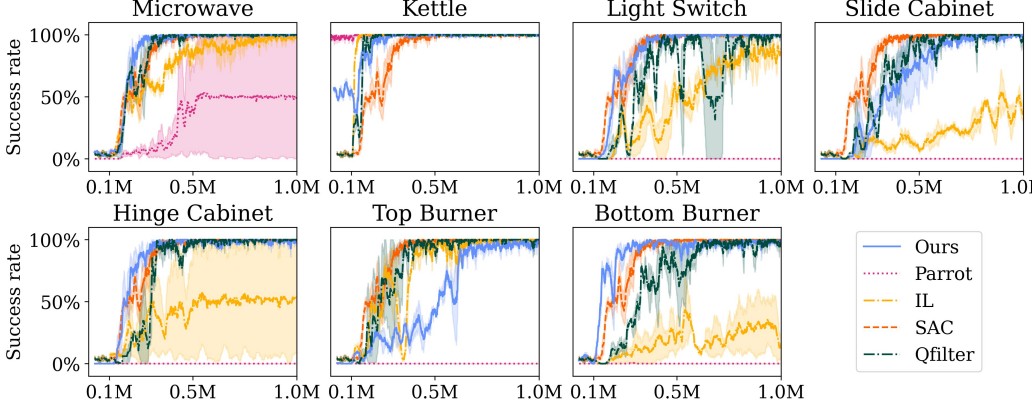

(b) Success rate over time corresponding to the above bar plot, using prior data $\mathcal{D}_j$ from different tasks (panel titles). The shaded area corresponds to one standard deviation across two random seeds.

Figure 13: Results on FrankaKitchen's `kettle` task, which requires sliding the kettle onto the stove burner.

# E   ADDITIONAL ABLATION RESULTS

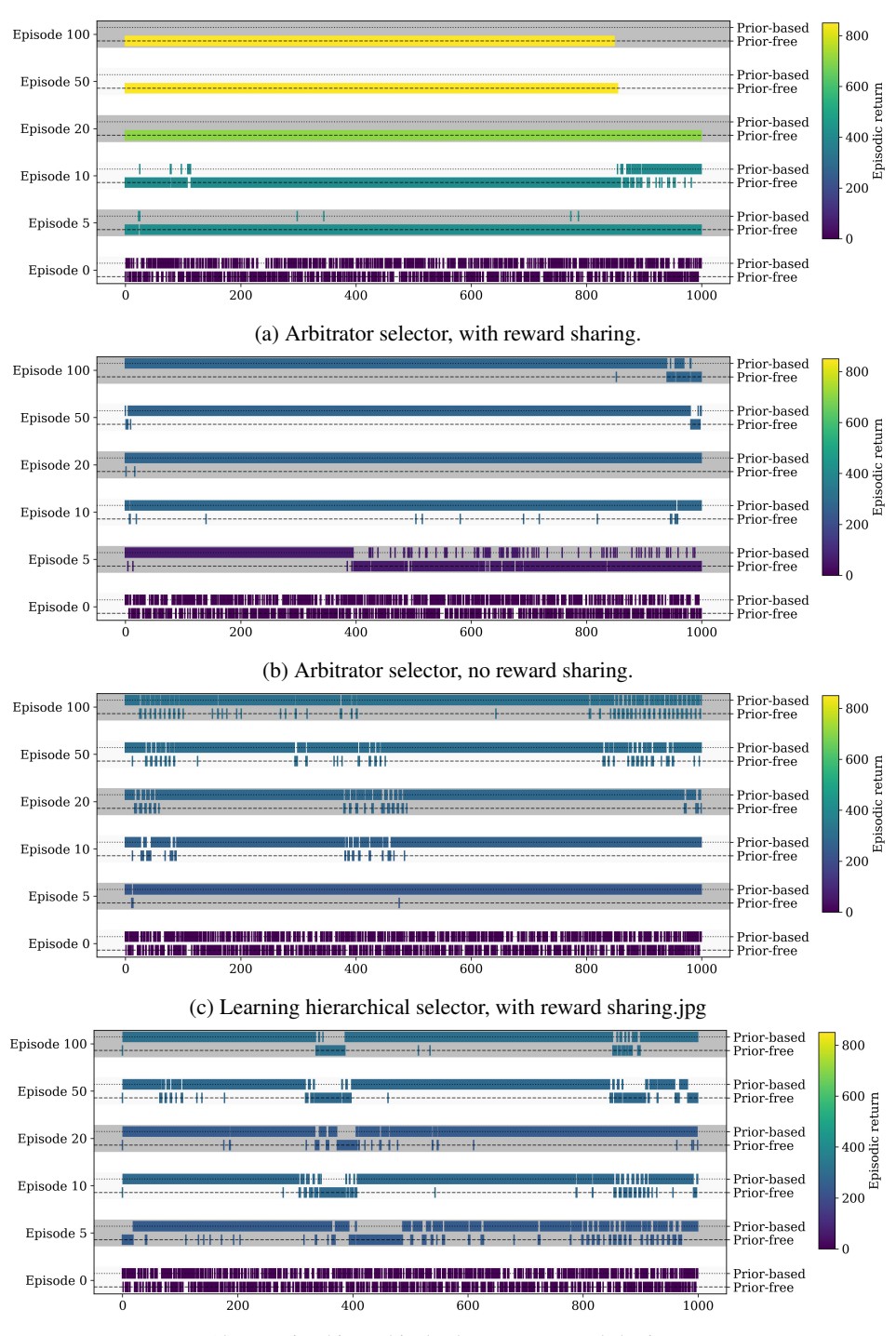

(a) Arbitrator selector, with reward sharing.

(b) Arbitrator selector, no reward sharing.

(c) Learning hierarchical selector, with reward sharing.jpg

(d) Learning hierarchical selector, no reward sharing.

Figure 14: Complete selector-action plot from the ablation study on the car racing environment, extension of Fig. 6.

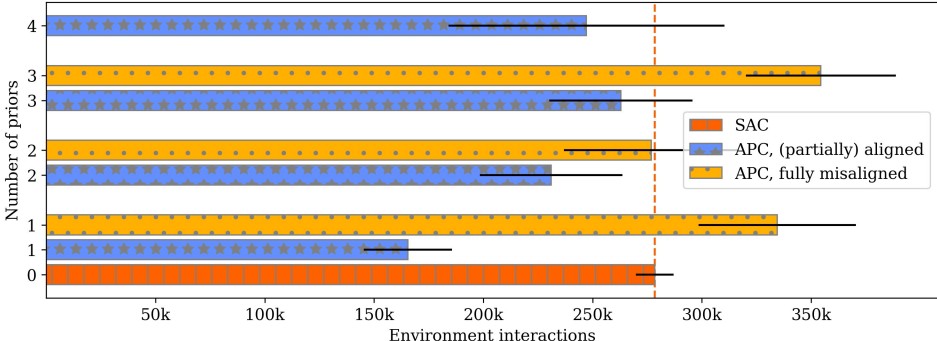

Figure 15: SAC vs APC's time until 100% cross-seed success, when using different numbers of behavior priors, on the `PointMaze` environment. The $x$-error bars indicates the variance over five seeds and the different tasks. APC's "(partially) aligned" variant here means that APC's set of priors includes the behavior prior for the current target task (but also $n - 1$ misaligned ones), while the "fully misaligned" variant means that the set of priors consists solely of *misaligned* priors for other tasks. The beneficial exploration bias due to the *aligned* prior is weakened by the increasing number of misaligned priors, which explains the increase in time until success when using more priors. Nevertheless, as long as APC has access to the aligned prior, it performs better or on par with SAC, highlighting its ability to avoid negative exploration bias from misaligned priors.

# F ADDITIONAL POINTMAZE RESULTS

## F.1 NUMBER OF PRIORS

We also study the effect of the number of misaligned priors on APC. For this, we run APC on the `PointMaze` environment with access to one, two, three, or all four behavior priors for the four goal locations. As before, we separately consider the performance when the set of priors is (partially) aligned, meaning it contains a behavior prior that is optimal for the current task, or fully misaligned, when the set of priors only contains behavior priors for other tasks.

For this analysis, we use a slightly denser reward for the `PointMaze` reward by replacing the standard exponent of the negative Euclidean distance reward with the negative Euclidean distance directly. The exponent of the negative Euclidean is uninformative, evaluating to 0 almost everywhere, except very close to the goal, effectively creating a sparse reward setting. In such settings, the effect of misaligned priors is amplified, since there is (almost) no feedback for learning about the negative influence of misaligned priors, allowing them to continuously bias the exploration in a negative fashion. Using the negative Euclidean distance directly yields more informative gradients throughout the maze, which allows us to better analyze how APC behaves as the number of priors increases.

The results are shown in Figure 15. As can be seen, with an increasing number of priors, the beneficial exploration bias from the aligned behavior prior is weakened. This happens because initially, before the Q-functions contain meaningful estimates, the misaligned behavior priors can (negatively) influence exploration by biasing the agents towards wrong goal locations. However, once the per-actor Q-functions correctly reflect lower values for the misaligned priors, they receive less weight under the arbitrator and their impact diminishes, allowing APC to exploit the aligned prior or prior-free actor.

Furthermore, we find that APC, when given a set of *fully* misaligned priors, performs slightly worse than from-scratch SAC, due to the misaligned priors biasing exploration negatively, until their Q-values reflect low utility. However, this represents a contrived adversarial setting that we include in this analysis for pedagogical reasons and completeness, but note that it is very unlikely in practice.

Figure 16: Arbitrator sensitivity analysis with respect to the Boltzmann temperature $\eta$. The arbitrator is largely insensitive towards the temperature, however, robustness in the misaligned prior case deteriorates with very high temperature $T = 100$.

## F.2 ARBITRATOR TEMPERATURE SENSITIVITY

We evaluate the sensitivity of the arbitrator to its temperature coefficient $\eta$. Recall that the arbitrator selects among actors using a Boltzmann categorical distribution $\pi_\beta = \mathrm{Cat}(p_0(\mathbf{s}), \dots, p_n(\mathbf{s}))$, with selection probabilities

$$p_l(\mathbf{s}) = \frac{1}{Z} \exp\left(\tfrac{1}{\eta} V^{(l)}(\mathbf{s})\right), \qquad Z = \sum_{i=0}^{n} \exp\left(\tfrac{1}{\eta} V^{(i)}(\mathbf{s})\right). \tag{13}$$

The temperature $\eta$ regulates how strongly value differences influence the selection: small $\eta$ yields a more peaked distribution, while large $\eta$ biases the distribution towards uniform. Figure 16 reports running success rates for temperatures on a logarithmic grid, $T \in 0.01, 0.1, \dots, 100$, evaluated on the PointMaze environment and using a single prior.

Overall, the arbitrator is mostly insensitive to the temperature choice. The only notable degradation appears for large temperatures and when given a misaligned prior, which is expected: In this setting, value differences between the misaligned prior-based actor and the prior-free actor remain too small relative to the temperature, leading the arbitrator to sample both nearly uniformly and preventing effective filtering of the misaligned prior.

A practical heuristic for choosing $\eta$ could be based on the reward scale: low reward magnitudes (and therefore small value differences) could be amplified with a smaller $\eta$, whereas high reward magnitudes might benefit form using a larger $\eta$, to counteract large value differences between even similar behaviors.

