# OpenReview forum: "APC-RL: Exceeding data-driven behavior priors with adaptive policy composition"
_ICLR.cc/2026/Conference — ICLR 2026 Poster_

### Official Review · Reviewer_LZVv · 2025-10-26

**Soundness:** 3
**Presentation:** 4
**Contribution:** 3
**Rating:** 6
**Confidence:** 3

**Summary:**

This work builds upon substantial advancements in behavioral priors for online reinforcement learning, and proposes an end-to-end technique to balance and control the guidance received from multiple, potentially misaligned priors. Given N data sources, the method trains N+1 policies: one acting over real actions, and N acting over the latent space of frozen normalizing flows estimating the state-conditional action distribution in each data source (as pioneered by PARROT). These policies, as well as their values, are simply trained through SAC; the softmax of the value can be used as a policy-selection criterion for online data collection. Finally, the connected data by each policy can be relabeled according to the transformations induced by each prior, and used for training every other policy. The authors evaluate the proposed methods on a simple 2D maze, as well as on simulated kitchen and driving environments. Across the board, results confirm that the proposed method is less sample efficient than PARROT when the prior is well aligned (likely due to the overhead of value learning for high-level action estimation), but prevent catastrophic failure when the prior is misaligned, while consistently outperforming a solution learning from scratch (as long as the prior is not adversarial). The submission finally provides ablations and a visualization of the policy switching pattern in the racing environment.

**Strengths:**

- The method is clear, principled and well motivated.
- The paper is well-written and easy to follow; claims are well supported and not overstated.
- The experimental section is well-designed and sufficient to analyze the most important facets of the algorithm. Notably, it includes both simulated and human-collected data.

**Weaknesses:**

- The computational overhead induced by the method is perhaps the biggest limitation: a linear number of agents need to be trained as the number of data sources increases. Have the authors considered conditional actor and critic architecture (i.e. a single critic network, conditioned on the policy's index)?
- A further constraint is the reliance of this method on invertible generative models in order to effectively relabel actions. Normalizing flows are in general capable models, but other methods (such as flow matching) seem to be prevalent empirically. Can the method be generalized to a broader class of architectures for modeling priors?
- The work fails to acknowledge that the proposed method may only alleviate partially misaligned priors. Consider a setting in which the agent is initialized in the top left corner of the maze, and the prior is trained on data reaching the top right corner, while the downstream task involves reaching the bottom left corner. Exploration, in this case, will be biased towards an useless direction, and it is hard to imagine that the method would outperform learning from scratch. Moreover, the normalizing flow trained on the prior would often be queried OOD. This is of course understandable, but I believe it should be acknowledged explicitly in a limitation section.
- There are several options for dynamically incorporating behavioral priors into downstream RL, including [1] and [2] (although I believe [2] does not directly apply as uncertainty estimates are not available nor informative). Although they have not been applied in the exact same setting, a comparison would be informative.

References:
[1] Bagatella et al., SFP: State-free Priors for Exploration in Off-Policy Reinforcement Learning, TMLR 2022
[2] Cramer et al., CHEQ-ing the Box: Safe Variable Impedance Learning for Robotic Polishing, arXiv 2025

**Questions:**

## Minor concerns and questions
- line 202: a definition and discussion of "primacy bias" might be informative.
- equation 4: I would recommend simply expressing the policy through $\propto$, and avoiding an explicit normalization constant.
- equation 4: did the authors also evaluate a greedy high-level policy? i.e. simply an argmax of the values
- line 218: did the authors actually measure high variance? Would more actions samples result in better empirical performance?
- line 244: why is a non-standard shape chosen for the D4RL maze.
- line 310: Fig.3 right instead of left
- Figure 5: how is the network-based high-level actor trained?
- line 716: this would be appreciated.

## Conclusion
This work is clearly motivated, the algorithm is principled and sufficiently supported by empirical evidence, particularly using both simulated and human-collected data. Despite computational and modeling limitations (as highlighted above), and assuming that questions are sufficiently addressed, I would currently recommend acceptance.

---

> ### Author Response · Authors · 2025-11-21
> **Part 1**
>
> Thank you for your detailed and thoughtful review. We very much appreciate your recognition of the paper’s **clear presentation**, the **principled design of APC**, and the **strong experimental evaluation**. Your comments raise highly relevant points, and we have incorporate the suggested clarifications and related discussions into the revised manuscript.
>
> # Regarding the computational demands
>
> Since other reviewers also commented on APC's computational overhead, we jointly reply to this point in our global response and kindly refer the reviewer there.
>
> # Regarding the need for invertible generative models
>
> APC indeed relies on invertible generative models, such as Normalizing Flows, to enable the reward-sharing trick via latent-space "relabeling". We explicitly designed APC to address the limitation observed in PARROT, where NF priors tend to remain overly restrictive under demonstration misalignment or distribution shift.
>
> While the overarching idea of combining a prior-free actor with one or more pre-trained behavior priors could be generalized to other generative models (e.g., flow-matching or diffusion models), this would require modifying or replacing the reward-sharing mechanism. We see this as a substantial deviation from APC with normalizing flows, and thus as promising future research direction, but out of scope for the current work.
>
> # Regarding fully misaligned priors
>
> This is an excellent and important observation, and we agree that it deserves acknowledgment in the revised manuscript. We have added this aspect to the new "Discussion and Limitations" section.
>
> It is correct that misaligned priors bias exploration toward sub-optimal regions of the state space. However, as each actor’s Q-function is updated, misaligned priors will propose actions with _lower_ values and are therefore selected increasingly rarely by the selector. As learning progresses, this negative bias diminishes, allowing APC to achieve similar asymptotic performance as a prior-free, from-scratch learning agent.
>
> That said, we agree that APC is unlikely to _outperform_ from-scratch learning in terms of exploration efficiency in a  fully misaligned prior setting, as exploration would be initially negatively biased. In practice, we observed that even misaligned demonstrations can _sometimes_ accelerate downstream learning, but this depends strongly on task semantics and the degree of behavioral similarity between priors and the target task, which is not straightforward to estimate a priori.
>
> # Regarding related works
>
> Thank you for pointing out these works. Both are indeed relevant, though we also see these approaches as somewhat orthogonal: CHEQ relies on an ensemble of critics to estimates agent uncertainty to fall back onto a control policy, while SFP aims to discover task _independent_ but temporally extended behavior for accelerating downstream exploration. Moreover, both of these methods integrate only a single prior with a single online RL agent, lacking APC's flexibility to integrate multiple distinct prior behaviors. We have incorporated this into the related works section.

---

> ### Author Response · Authors · 2025-11-21
> **Part 2**
>
> # Minor concerns and questions
> + _202: A definition and discussion of "primacy bias" might be informative._ Agreed, we've added the definition and reference pointers to relevant works in the revised version.
> + _Equation 4._ We find the explicit normalization constant clearer, but we agree that using "$\propto$" would also work. We experimented with a greedy ($\arg\max$) selection instead of the softmax during early implementations but found that the softmax formulation was more robust overall.
> + _line 218: Would more action samples result in better empirical performance?_ Yes, increasing the number of samples would reduce the variance of the value-function estimate and could be integrated straightforwardly. We primarily used the single-sample Monte Carlo approximation for algorithmic simplicity.
> + _line 244: why is a non-standard shape chosen for the D4RL maze?_ We created a custom maze because we wanted to create a layout where exploration under misaligned priors becomes particularly challenging, in order to highlight the issues that arise when such priors are rigidly integrated.
> + _line 310: Fig.3 right instead of left._ Thank you for spotting this, corrected in the revision!
> + _Figure 5: how is the network-based high-level actor trained?_  We appreciate the opportunity to clarify and acknowledge that this was not clearly outlined in the initial manuscript -- we have added the following to the revised version: The hierarchical selector is implemented as a SAC agent trained directly to maximize online reward. Its action space is discrete, the actor parameterizes a Categorical distribution representing the choice over which lower-level actor to execute. Thus, at each step $t$, the network-based high-level actor estimates which latent actor will achieve the highest expected return and executes that actor.
> + _line 716: [completing the Franka Kitchen benchmark] would be appreciated._ We have completed the benchmark in the meantime, results are in the appendix of the revision and are consistent with those in the initial version of the paper.

---

> > ### Comment · Reviewer_LZVv · 2025-11-22
> >
> > Thank you for answering my questions. Computational and modeling limitation still hold, but are openly acknowledged. I am happy to keep my score and recommend acceptance.

---

> > > ### Author Response · Authors · 2025-11-24
> > >
> > > Thank you very much for your engagement, feedback, and support!

---

### Official Review · Reviewer_fyEH · 2025-10-31

**Soundness:** 2
**Presentation:** 3
**Contribution:** 2
**Rating:** 2
**Confidence:** 5

**Summary:**

The paper proposes APC-RL, which combines multiple prior-based actors (using normalizing flows) with a prior-free actor under a value-based selector. It introduces a “reward-sharing” trick that reuses experience across actors via NF invertibility. Experiments on PointMaze, FrankaKitchen, and CarRacing claim faster learning and robustness to misaligned demonstrations.

**Strengths:**

1. Tackles a relevant problem: overcoming poor demonstrations in RL.

2. Simple and modular architecture that integrates easily with SAC.

3. Reward-sharing via NF inversion is conceptually neat and potentially sample-efficient.

**Weaknesses:**

1. Efficiency and justification gaps: The approach may be inefficient in multiple ways: maintaining several actors, evaluating each at every step, and managing multiple replay buffers. The paper neither analyzes this computational overhead nor justifies why keeping multiple actors is preferable to simply retraining a single policy once demonstrations start degrading performance.

2. Limited generality: The method is evaluated only with SAC, with no discussion of applicability to other off-policy algorithms.

3. Invertibility not clearly needed: Since SAC is off-policy, rewards could be shared directly without NF inversion. The necessity of invertibility is not well justified.

4. No clear intuition: The paper details how training occurs but not why the composition improves learning or stability.

5. Weak experimental design: Few baselines, undefined notion of “misalignment,” and only 2–3 seeds. The paper should also compare against established demonstration-filtering methods such as HER with demonstrations (Q-filter) to contextualize the claimed benefits.

**Questions:**

1. How is demonstrator misalignment quantified?

2. Why not directly share rewards without invertibility if the training is off-policy?

3. How efficient is the selector? Does it scale with many actors?

4. Would this method still outperform if retraining from scratch without demos?

5. How would APC-RL compare to HER with Q-filter on similar tasks?

---

> ### Author Response · Authors · 2025-11-21
> **Part 1**
>
> Thank you for your evaluation of our work and for acknowledging its **practical relevance**, the **simplicity of APC's integration**, and the **sample efficiency** achieved through our reward-sharing mechanism. We appreciate the opportunity to clarify the raised concerns, and we have addressed these in the revised manuscript, partially in a new *“Discussion and Limitations”* section.
>
> # Regarding APC's justification and computational demands
>
> Since other reviewers also commented on APC's computational overhead, we kindly refer the reviewer to our global response, where we comment on this once and jointly.
>
> Regarding the justification for multiple actors, we note that the separation between prior-based and prior-free actors is both intentional and essential. It allows us to preserve the utility of multiple pre-trained and distinct behavior priors without risking degradation and collapse through online updates, avoid performance ceilings due to distribution shift and initialization, and critically, enables the prior-free actor to diverge when demonstrations are misaligned. Our experiments demonstrate that this separation is precisely what ensures robustness and adaptability under misalignment.
>
> # Regarding off-policy generality
>
> Our current evaluation is consistent with that in PARROT and the general field, and focuses on SAC as a representative SOTA off-policy algorithm. But we agree that a brief discussion on APC's broader applicability would strengthen the manuscript: APC is conceptually independent of the underlying off-policy algorithm since we only assume off-policy replay buffers. Thus, APC can be readily combined with other off-policy algorithms such as DrQ, REDQ, or TD7 without modification.
>
> We have clarified the applicability of APC to other off-policy RL methods in the revised manuscript and would welcome further clarification from the reviewer if there are **specific and explicit** concerns about APC's compatibility with other off-policy RL algorithms.
>
> # Regarding the need for invertibility
>
> As we discuss in Section 3.1 (Compositional Policy Model), each actor operates in its **own latent space**, defined by its own Normalizing Flow (NF). These latent spaces are *not shared*, meaning we have $\mathcal{Z}^{(i)} \neq \mathcal{Z}^{(j)}$. When actor $i$ is selected at time $t$, a latent action $\mathbf{z}^{(i)}_t$ is mapped to an environment action via the corresponding NF transformation $T^{(i)}$ $a_t = T^{(i)}(\mathbf{z}^{(i)}_t; \mathbf{s}_t)$.
>
> But to update **all other latent actors**, with latent spaces $\mathcal{Z}^{(j\ne i)}$, APC must map the *same environment action* $\mathbf{a}_t$ back into **each actor’s distinct latent space**, i.e.  $\mathbf{z}^{(i)}_t \ne \mathbf{z}^{(j)}_t = \tilde{T}^{(j)}(\mathbf{a}_t; \mathbf{s}_t)$,  which, crucially, produces *different* latent coordinates for the different latent actors. This inverse mapping step is therefore a strict requirement for the reward-sharing mechanism. Without invertibility, these mappings cannot be computed, and latent actions from one actor would be invalid for updating other latent policies.
>
> Thus, contrary to the reviewer’s claim, off-policy RL alone is *not* sufficient for reward sharing when each policy acts in a different, learned latent space. This makes the invertibility of NFs a fundamental condition for APC and not a unjustified design choice.
>
> # Regarding the intuition
>
> We appreciate the opportunity to clarify the intuition behind why the compositional design of APC improves learning and stability. Our core design consideration is that **not all demonstration datasets are equally useful**: Some may be optimal, others might be suboptimal, and some might be actively misleading for the target task. Each prior-based actor in APC is initialized with the behavior in its own demonstration dataset and then refined through online RL. However, as shown in our work, NF behavior priors lack the flexibility to deviate fully from their demonstration data, if they are misaligned with the target task.
>
> APC’s compositional structure directly addresses this. The selector continuously compares the value estimates of all actors and adaptively decides which actor to rely on. Useful priors are exploited and refined; while unhelpful or misleading priors are down-weighted and eventually discarded. Crucially, this design also facilitates the inclusion of the additional prior-free actor, which ensures that even if *all* priors are poor or misaligned, APC can still learn without being constrained by the misaligned priors and data.
>
> So the inuition is to capture distinct behavior priors based on multiple demonstration datasets (that may be misaligned with the target task), and to adaptively compose and augment them based on online feedback, instead of regularizing a single online policy to include all of the demonstrated behaviors.

---

> ### Author Response · Authors · 2025-11-21
> **Part 2**
>
> # Regarding the experimental design
>
> The baselines in our evaluation were carefully selected for relevance and represent a broad spectrum of approaches. The imitation learning (IL) baseline is representative of KL or MSE-regularized imitation losses, the most common mechanism for integrating demonstrations, while PARROT serves as both our direct predecessor and an example of latent skill pre-training approaches that, unlike APC, lack the flexibility to sidestep misaligned priors. Together, these baselines capture the key methodological families relevant to our setting. Also note that other reviewers described our experimental evaluation as **comprehensive, convincing, and well-designed**.
>
> However we thank the reviewer for pointing out the HER + Q-filter method [1]. While HER is applicable only to goal-conditioned MDPs, a restriction not shared by APC, we agree that the Q-filter mechanism is conceptually relevant. We have implemented this baseline and already completed experiments on PointMaze and CarRacing, with FrankaKitchen results still pending. Results so far show that Q-filter can mitigate some negative effects of misaligned demonstrations; however, it is substantially less robust and less sample-efficient than APC.
>
> We hypothesize that this is due Q-filter relying solely on accurate Q-value estimates to distinguish useful from harmful $(\mathbf{s}, \mathbf{a})$ demonstration. Early in training, these Q-function estimates are unreliable, which limits both the ability to filter bad demonstration samples and the ability to exploit good ones. Furthermore, Q-filter requires the policy to imitate all beneficial behaviors in the dataset, which can be challenging when demonstrations are multimodal but the policy class (e.g., Gaussian) is unimodal. APC avoids these limitations. It can sample from rich, multimodal NF behavior priors immediately from the start, while being able to down-weight and side-step misaligned ones as indicated by the their Q-functions.
>
> We have included the Q-filter baseline results available so far into the revision and thank the reviewer again for the suggestion.
>
> [1] Overcoming Exploration in Reinforcement Learning with Demonstrations, Nair et al, 2018.
>
> ## Regarding the notion of misalignment
>
> We appreciate the opportunity to clarify the concept of *demonstration misalignment*, which is central to the motivation for our work, while also noting that reviewers add8, TwB7, and LZVv already find our paper clearly written and easy to follow. While we did not formalize it mathematically in the main text, the notion is conceptually straightforward to grasp: By *demonstration misalignment*, we refer to situations where the provided demonstrations are, in some way, suboptimal for the target task. For example, if the target task requires moving to a goal on the **left**, while the demonstrations only show trajectories towards the **right**, or if the demonstrations induce a **performance ceiling** (as in the CarRacing experiments), we say that such demonstrations are misaligned.
>
> Many prior methods, including PARROT and imitation-learning baselines, tend to implicitly assume that demonstrations are optimal and fully aligned with the online task (e.g. Fig. 7 in PARROT paper [1]), which leads to degradation when this assumption is violated. Our experiments explicitly demonstrate this failure mode and show how APC remains robust under such misalignment by adaptively bypassing misaligned priors, if needed.
>
> In principle, demonstration misalignment could be quantified if one had access to the optimal policy $\pi^*$, for example as using the Kullback–Leibler divergence $D_\text{KL}$
>
> $$\int_{\mathbf{s}} D_{\mathrm{KL}}(\mathcal{A}_{\mathcal{D}}(\mathbf{s}) || \pi^*(\cdot\mid\mathbf{s}))\, d\mathbf{s},$$
>
> where $\mathcal{A}_{\mathcal{D}}(\mathbf{s})$ denotes the empirical action distribution from demonstrations $\mathcal{D}$ and $\pi^* (\cdot | \mathbf{s})$ is the optimal action distribution at state $\mathbf{s}$. This integral would capture the per-state divergence between the demonstrated and optimal action distributions. Of course, in practice $\pi^*$ is unknown a-priori, which is why we study robustness empirically rather than quantifying and defining misalignment mathematically.
>
> [1] Parrot: Data-Driven Behavioral Priors for Reinforcement Learning, Singh et al, ICLR 2021 (Oral)
>
> # Questions
> > How is demonstrator misalignment quantified?
>
>  See above answer "Regarding the notion of misalignment"
>
> > Why not directly share rewards without invertibility if the training is off-policy?
>
> See above answer "Regarding the need for invertibility"

---

> ### Author Response · Authors · 2025-11-21
> **Part 3**
>
> > How efficient is the selector? Does it scale with many actors?
>
> Our selector is a **learning-free Boltzmann categorical** over per-actor value estimates
> $$
> p_l(s) \propto \exp\left(\tfrac{1}{\eta}\,V^{(l)}(s)\right),
> $$
> where in practice $V^{(l)}(s)$ is approximated with a single sample from each actor. This implies a number of Q-function evaluations **linear** in the number of used actors. This design keeps the selector's compute and memory footprint negligible and scales well to even large number of actors, while avoiding the well-known non-stationarity issues of learned hierarchical networks. We therefore don't see a scaling issues in the selector.
>
> > Would this method still outperform if retraining from scratch without demos?
>
> We may have misunderstood the reviewer's question, but if it refers to APC's performance when not given *any* demonstration data at all, then APC simply reduces to standard SAC: Without any demonstration datasets, there are no prior actors to pre-train and to include, so APC solely includes the prior-free actor, which will be selected and updated at every step, since the arbitrator's Boltzmann categorical distribution becomes a degenerate distribution with just one outcome. Thus, when APC is not given any demonstration data, it reduces to standard SAC (or whichever implementing off-policy algorithm is chosen), with no reason for it to out- or under-perform that method.
>
> > How would APC-RL compare to HER with Q-filter on similar tasks?
>
> Please see our above response "Regarding the experimental design"

---

> ### Author Response · Authors · 2025-11-25
> **Part 4**
>
> We would like to inform the reviewer that we have now uploaded a revision that contains the full results for the Q-filter baseline. The previous revision only had Q-Filter results for the CarRacing and PointMaze environments, now we also have the complete FrankaKitchen benchmark (see Figure 4 in main text and Section D in the appendix).
>
> Across all our experiments, we find that **APC consistently outperforms Q-Filter**, meaning that APC demonstrates better sample efficiency and performance than Q-filter when using both **aligned demonstrations or misaligned demonstrations**, across all of our tested environments. This behavior can be well explained and justified by Q-Filter's and APC's underlying architectures: APC can immediately leverage the pre-trained, distinct, and potentially multimodal behavior priors for exploration, whereas Q-filter can only exploit (or ignore) demonstrations once the Q-function reliably and accurate reflects their quality. In addition, Q-filter requires the policy to learn to imitate all beneficial demonstration behaviors, which in itself can be difficult when the dataset is multimodal but the policy class (e.g., Gaussian) is unimodal.
>
> We hope these results address the reviewer’s question regarding APC’s performance relative to Q-filter. We again thank the reviewer for suggesting this relevant baseline, we believe this addition has meaningfully strengthened our manuscript.

---

> ### Author Response · Authors · 2025-11-27
>
> Dear Reviewer,
>
> Thank you again for your valuable time and feedback. We completely understand that the discussion phase coincides with many competing commitments. As the author-reviewer discussion period now enters its final week, we would be very grateful if you could take a moment to review our earlier responses and let us know whether they adequately address your concerns. If any points remain unclear, or if additional questions have come up after reading our replies, we would be happy to clarify them while the discussion phase is still open.
>
> We sincerely appreciate your time and consideration.

---

> ### Author Response · Authors · 2025-12-01
> **Concluding Statement for AC**
>
> Dear AC,
>
> Due to the premature termination of the discussion period, we would like to provide a final statement and clarification regarding the review from reviewer feYH.
>
> Firstly, reviewer feYH appears to have misunderstood a central aspect of our method, namely that each normalizing-flow prior operates in a different latent space. We believe this misunderstanding directly gave rise to their critique "Weakness 3: Invertibility not needed". During the rebuttal we provided a detailed explanation and made this aspect of our method even clearer in the revised manuscript. In contrast, **all other reviewers correctly understood this aspect of our method** from the initial manuscript and described the paper as "clear" and "easy to follow". This contrast makes the reviewer's confidence score of 5 somewhat unfortunate.
>
> Additional claims by reviewer feYH (e.g., "weak experimental design", "no clear intuition") are also **inconsistent with the feedback from the other three reviewers**, who independently describe the paper as "well-motivated" and "clear", and the experiments as "high-quality", "convincing", and "thorough". This contrast suggests low engagement and possibly a predisposition towards rejection from reviewer feYH.
>
> Lastly, reviewer feYH requested an additional and relevant baseline (Q-Filter). We have included it in the revision and show that **our method outperforms it across all benchmarks**. We also explain why this occurs, both in the paper and in our rebuttal message to the reviewer.
>
> We respectfully ask that this context is taken into consideration when synthesizing the final decision.
>
> Thank you for your time and consideration.

---

### Official Review · Reviewer_TwB7 · 2025-11-01

**Soundness:** 3
**Presentation:** 3
**Contribution:** 3
**Rating:** 6
**Confidence:** 3

**Summary:**

This paper introduces Adaptive Policy Composition (APC), a framework that learns multiple data-driven Normalizing Flow (NF) priors from different demonstration datasets. Building on these NF priors, the method trains multiple latent policies and corresponding low-level actors in online environments. A reward-sharing trick leverages the invertibility of NF priors, allowing different actors to learn from the same transition. A high-level selector is then learned to determine which low-level actor to invoke at each state. The proposed approach effectively addresses the challenge of suboptimal demonstration data in real-world scenarios. Experimental results show that APC outperforms the PARROT baseline.

**Strengths:**

1. The reward-sharing trick is particularly interesting, as it allows different actors to learn from a single data source, thereby improving sample efficiency.

2. The paper presents comprehensive experimental results across multiple environments, demonstrating the effectiveness of the proposed method.

**Weaknesses:**

1. Since the proposed APC method learns multiple actors from the environment, it may incur additional computational costs compared to traditional methods (e.g., PARROT). Moreover, the approach requires multiple demonstration datasets, which may not be applicable in real-world scenarios.

2. The ablation study in the paper is also not sufficiently strong. For instance, it lacks an analysis of performance under limited demonstration sources (e.g., using only a single dataset) and does not include comparisons with state-of-the-art Learning-from-Demonstration (LfD) methods using optimal or suboptimal data.

**Questions:**

1. If I understand correctly, the authors use the selector to choose a single actor at each step rather than forming a mixture of multiple actors. How would the performance be affected if the method instead used a mixture of different actors? If such a mixture could hinder actor training, could it still be beneficial after the actors have been trained?

2. What is the main difference between APC and existing methods that rely on a single data source? If other Learning-from-Demonstration (LfD) methods were extended to integrate the multiple demonstration datasets used in this work, would they achieve similar performance?

3. From Figures 3(b) and 4(b), the proposed APC method does not show substantial improvement over PARROT or SAC on aligned data (with slower convergence). Can I assume that the method mainly provides advantages when the demonstration data are misaligned?

4. I am interested in the performance of APC on the D4RL benchmarks. How does it compare with state-of-the-art LfD methods trained on datasets with different levels of expertise? Would APC still outperform those methods if only a single expert demonstration dataset were used?

---

> ### Author Response · Authors · 2025-11-21
> **Part 1**
>
> Thank you for your constructive and insightful review. We are pleased that you found our **empirical study comprehensive** and our **reward-sharing mechanism effective** for improving sample efficiency. We appreciate the opportunity to clarify parts of our method based on the brought forward critiques and questions and have integrated relevant points of the discussion into the revised manuscript.
>
> # Regarding the computational demands
> Since other reviewers also commented on APC's computational overhead, we jointly address to this point in our global response and kindly refer the reviewer there.
>
> # Regarding the ablation study
> We agree that understanding APC's behavior under limited (single-task) data is important. However, our CarRacing experiment is meant to directly address this setting, as it uses a single suboptimal demonstration dataset collected from a human driver. The results (Fig. 5) show that APC not only outperforms from-scratch SAC but also exceeds both imitation learning and PARROT, demonstrating APC's ability to exploit limited, suboptimal data while avoiding the performance ceiling that affects existing, less flexible LfD approaches.
>
> Moreover, all our experiments include comparisons with what we consider to be representative LfD baselines: The "IL" baseline captures methods that regularize toward demonstrated actions (e.g. KL or MSE-based BC policy loss terms), while PARROT represents approaches that rely on pre-training latent "skill" spaces. We believe that policy-regularization and generative, latent-skill-based approaches cover many LfD methods.
>
> If the reviewer had a specific LfD method in mind that differs conceptually from our baselines, we would appreciate any pointers.
>
> # Questions
>
> > How would the performance be affected if the method instead used a mixture of different actors?
>
> Thank you for the point and suggestion. APC currently selects a single actor at each step, but it could in principle be extended to learn or sample from a mixture of the actors, potentially using the Q-values we currently use for sampling as mixture weights. Such a mixture could improve exploration efficiency, since it facilitates complex behaviors by blending the separate actors.
>
> While we do not expect the asymptotic performance to change _substantially_ (since we can always achieve the asymptotic performance of SAC with the prior-free actor), a mixture would improve the expressivity of the policy distribution and might, in some MDPs, yield even better performance.
>
> We see this as a promising and exciting direction and have explicitly mentioned it in the revised manuscript as valuable future work.
>
> > What is the main difference between APC and existing methods that rely on a single data source? If other Learning-from-Demonstration (LfD) methods were extended to integrate the multiple demonstration datasets used in this work, would they achieve similar performance?
>
> We appreciate the opportunity to clarify the distinction between APC and standard LfD approaches. We have integrated this into related works section of the revised manuscript.
>
> The key distinction between APC and standard LfD approaches lies in APC's inclusion of a prior-free actor and the adaptive selection mechanism, which together allow it to learn when and where to rely on demonstration priors, and when to diverge from them, to achieve optimal performance. Standard LfD methods typically implement BC losses, either via KL or MSE, and lack the flexibility of APC. They enforce imitation of the demonstration data regardless of its task relevance, which can severely degrade performance when demonstrations are sparse, mixed, or in some other way misaligned, as our experiments show. APC therefore does more than simply integrating behavior priors from multiple demonstration datasets, explaining the observed increased performance under demonstration misalignment.
>
> Moreover, APC maintains an advantage over LfD methods that rely solely on demonstration *data* rather than demonstration *priors*. Because APC can immediately sample from a rich, NF behavior prior, it can exploit **multimodal** expert behaviors from the very beginning of training, without first needing to learn these behaviors into a potentially **unimodal** (e.g., Gaussian) policy. This can allow for more efficient exploration with APC, even in the aligned single-dataset setting.

---

> ### Author Response · Authors · 2025-11-21
> **Part 2**
>
> > Can I assume that the method mainly provides advantages when the demonstration data are misaligned?
>
> Yes, the main advantage of APC over PARROT (or other IL/BC methods) lies in its ability to efficiently exploit, but also diverge, from a number of given behavior priors. When the given prior(s) perfectly match the task, that is if pure behavior cloning achieves optimal performance, then there is not much to be gained from APC's flexibility. In such cases, we can expect APC to perform on par or slightly better than other demonstration guided methods (due to the advantages mentioned before), and to outperform in terms of sample efficiency compared to learning from scratch with, e.g., SAC.
>
> > I am interested in the performance of APC on the D4RL benchmarks.
>
> We appreciate the reviewer’s interest in this direction. The complete D4RL benchmark would indeed provide a valuable testbed. However, we are not aware of existing D4RL variants that include misaligned or mixed demonstrations, which are central to evaluating APC's main advantage,  its robustness under demonstration misalignment.
>
> Evaluating APC on D4RL tasks with varying levels of expertise would be a compelling extension to our experimental testbed, we agree. Unfortunately this lies beyond the scope of what we can do during the rebuttal.
>
> When given a fully aligned, single expert demonstration dataset, we expect APC to perform on par with existing demonstration-guided methods. APC’s primary advantages are flexibility and robustness under misalignment, but these are not needed if we can optimally solve the task by doing, essentially, pure behavior cloning. In this case, a prior-based actor would effectively provides a perfect initialization, and the selector would quickly settle on and exploit this. Thus, we do not expect APC to outperform PARROT in this specific case.

---

> ### Author Response · Authors · 2025-11-27
>
> Dear Reviewer,
>
> Thank you again for your valuable time and feedback. We completely understand that the discussion phase coincides with many competing commitments. As the author-reviewer discussion period now enters its final week, we would be very grateful if you could take a moment to review our earlier responses and let us know whether they adequately address your concerns. If any points remain unclear, or if additional questions have come up after reading our replies, we would be happy to clarify them while the discussion phase is still open.
>
> We sincerely appreciate your time and consideration.

---

### Official Review · Reviewer_add8 · 2025-11-01

**Soundness:** 3
**Presentation:** 4
**Contribution:** 3
**Rating:** 6
**Confidence:** 3

**Summary:**

The authors propose a hierarchical RL approach to leveraging suboptimal/misaligned demonstration data; specifically, they propose Adaptive Policy Composition (APC), which trains multiple Normalizing Flow (NF) prior policies and chooses one based on alignment with the target task. Crucially, APC uses two key mechanisms to maintain performance in the face of non-optimal / misaligned demonstrations: (1) reward-sharing, which exploits NF invertibility to map each action token to all policies' latent spaces, enabling continuous updates across all policies and not just the one which took the action, (2) a parameter-free "arbitrator" high-level selector, which simply selects the policy based on the value estimate of the lower-level actors.

This method surpasses PARROT [1] as well as standard from-scratch RL and imitation learning baselines in the misaligned and suboptimal demonstration settings and is only slightly slower to converge compared to PARROT given a fully-aligned prior. Ablations demonstrate that both mechanisms are critical for performance: the arbitrator avoids overcommitting to the prior-based actor and maintains balanced usage of actors, and reward-sharing ensures all actors are updated continuously.

[1] Singh, Avi, et al. "Parrot: Data-driven behavioral priors for reinforcement learning." arXiv preprint arXiv:2011.10024 (2020).

**Strengths:**

Leveraging suboptimal / misaligned demonstration data for reinforcement learning is a significant research topic, given that real-world scenarios are unlikely to have perfectly aligned demonstration data.

The proposed method is a novel contribution; in particular, the reward-sharing trick which exploits invertibility of normalizing flows which enables continuous updates for all actors.

The experimental section is executed thoroughly and with high quality; the ablations are convincing that both mechanisms are critical for final performance. Overall the paper is well-motivated and design choices at each step is clear.

**Weaknesses:**

APC’s framework trains multiple parallel SAC agents and evaluates each, so (as typical for methods that target the low-sample regime) while the sample-efficiency is increased, the total compute and (likely) wall-clock time may actually increase. Clarifying the scale of this extra compute (and e.g. memory for the additional replay buffers) per task suite could be helpful.

Besides Franka Kitchen, evaluation settings are relatively simple (Maze Navigation and Car Racing); demonstrating this works on additional task suites would be much more convincing.

**Questions:**

Could you elaborate on how to get the Normalizing Flow reward-sharing method to work for discrete action spaces?

Performing some sensitivity analysis on the temperature parameter for the "parameter-free" arbitrator (which is therefore a bit of a misnomer) could be useful as well.

Are there any pathological situations in which, due to the distribution of demonstration data, APC would learn significantly slower than the learning-from-scratch baseline, or is there an argument for why this cannot occur?

It would be clearer to explicitly state the number of actors per evaluation environment (e.g. one of them only had a single prior-based actor), and it would be very interesting to find the effect on performance as the number of actors (and therefore demonstration datasets) scaled. Could you have number of actors != number of datasets (e.g. suppose the number of tasks is extremely high)?

---

> ### Author Response · Authors · 2025-11-21
> **Part 1**
>
> Thank you for your thorough review of our work. We appreciate your acknowledgment of the **novelty and relevance** of our method, as well as the **quality and thoroughness** of our experiments. Your brought forward points are constructive and clarifying, and we have incorporated them into the revised manuscript.
>
> # Regarding the computational demands
> Since other reviewers also commented on APC's computational overhead, we jointly reply to this point in our global response and kindly refer the reviewer there.
>
> # Regarding the evaluation settings
> We agree that both the 2D Maze and CarRacing environments are relatively simple. However, they are sufficient to clearly expose the limitations of prior methods and to demonstrate that all components of APC are necessary. The results consistently show that APC outperforms from-scratch learning when demonstrations are aligned, while also outperforming demonstration-guided baselines under misalignment. We thus don't see additional evaluations necessary for supporting our claims, but agree that further benchmarks would strengthed our interpretation even more.
>
> # Questions
>
> > Could you elaborate on how to get the Normalizing Flow reward-sharing method to work for discrete action spaces?
>
> To the best of our knowledge, NF priors for RL, as originally introduced in the PARROT paper, are only applicable to MDPs with continuous action spaces. This is because NFs require invertible, deterministic, and differentiable transformations, which can't straightforwardly model categorical target distributions over environment actions. Though there are a small number of works on discrete NFs, they differ conceptually and methodologically from the way NFs are used as behavior priors for RL. We will mention the integration of discrete NF behavior priors as promising future work direction.
>
> > Performing some sensitivity analysis on the temperature parameter for the "parameter-free" arbitrator (which is therefore a bit of a misnomer) could be useful as well.
>
> We appreciate the observation regarding our description of the arbitrator as "parameter-free". We agree that it does, in fact, include one parameter, the temperature coefficient $\beta$. Our intention was to highlight that the arbitrator does not require _learning_ any additional parameters, as its decisions are computed directly from the Q-values of the lower-level actors. We have revised the terminology and now refer to the arbitrator as *learning*-free, rather than parameter-free. Thank you for pointing this out.
>
> As suggested, we have performed a sensitivity analysis on the Boltzmann temperature and included it under Appendix F in the revision. The results show little to no sensitivity to the temperature on reasonable scales, but also that a _very large_ temperature can lead to performance degradation, since it essentially converts the arbiter informed by Q-values into a uniform distribution.

---

> ### Author Response · Authors · 2025-11-21
> **Part 2**
>
> > Are there any pathological situations in which, due to the distribution of demonstration data, APC would learn significantly slower than the learning-from-scratch baseline, or is there an argument for why this cannot occur?
>
> A very interesting point. Indeed, it is possible to construct *adversarial* theoretical settings in which APC could perform worse than from-scratch learning.
>
> Consider a case with $n$ strongly misaligned priors, each biasing exploration toward task-irrelevant or counterproductive behaviors. Assume a sparse-reward environment and a trajectory length of $T$. For simplicity, let us consider a discrete action space, where the probability of selecting the optimal action at any step under uniform exploration is $\frac{1}{|\mathcal{A}|}$.
>
> For standard from-scratch learning, the probability of sampling the optimal trajectory purely by chance is therefore $\left(\frac{1}{|\mathcal{A}|}\right)^T$. For APC, if all $n$ priors are misaligned, the only actor that _could_ propduce the optimal action is the prior free one, and the probability of selecting the that actor at any step would be $\frac{1}{n}$. Consequently, the probability of executing the optimal trajectory with APC would be $$\left(\frac{1}{|\mathcal{A}|} \times \frac{1}{n}\right)^T = \frac{1}{n^T |\mathcal{A}|^T},$$ which is indeed lower than the from-scratch case. In other words, in a sparse-reward setting with many misleading priors, exploration could become less effective because the prior-free actor, the only one capable of exploring and learning the optimal policy, would be selected less frequently.
>
> In *dense-reward* settings, however, this effect would be far less severe. Misaligned priors would receive low returns, leading to low estimated Q-values. Under the Boltzmann arbitrator, their selection probability would quickly diminish, and APC would naturally shift exploration toward the prior-free actor. As a result, APC effectively falls back to from-scratch learning after a short, initial adaptation period.
>
> Thus, while it is theoretically possible for APC to learn more slowly than from-scratch RL in contrived adversarial conditions with many strongly misaligned priors and sparse rewards, this only affects exploration and not asymptotic performance. Moreover, any demonstration-guided method that integrates priors more rigidly, such as through KL regularization or hard behavioral constraints, would collapse entirely in such a scenario, whereas APC retains the flexibility to recover and learn effectively.
>
> We will integrate a concise version of this discussion into the revised manuscript under a new "Discussions and Limitations" sections and thank the reviewer for raising this insightful point.
>
> > [...] it would be very interesting to find the effect on performance as the number of actors (and therefore demonstration datasets) scaled. Could you have number of actors != number of datasets (e.g. suppose the number of tasks is extremely high)?
>
> We agree that the effect of the number of priors is highly relevant and thank the reviewer for raising this point. We have therefore conducted an additional evaluation on the 2D Maze environment to investigate exactly this aspect (included in the revision under Appendix F).  There, we find that APC's asymptotic performance remains essentially independent of the number of priors used. However, the exploration efficiency decreases slightly as the number of _misaligned_ priors increases. This behavior is expected: each misaligned actor temporarily biases exploration toward suboptimal behavior, until the learned Q-values correctly reflect lower utility for those misaligned actors.
>
> And yes, APC absolutely supports integrating a number of actors $\ne$ number of datasets. Generally speaking, if we have multi-task, diverse demonstration data, and if we have knowledge about how to split this data into meaningful and distinct behaviors, doing this is beneficial because it gives raise to specialized, higher-quality behaviors priors, comparred to just one prior actor that attempts to perform the mean behavior observed in the dataset. But if we don't have knowledge how to split the demonstration dataset (or if the number of datasets would be very large), from APC's perspective there is no problem with training just one behavior prior, and augmenting that one prior with APC. APC would simply optimize the prior with online feedback by learning in that prior's latent space, and rely on the prior-free actor as needed for achieving optimal behavior.

---

> ### Author Response · Authors · 2025-11-27
>
> Dear Reviewer,
>
> Thank you again for your valuable time and feedback. We completely understand that the discussion phase coincides with many competing commitments. As the author-reviewer discussion period now enters its final week, we would be very grateful if you could take a moment to review our earlier responses and let us know whether they adequately address your concerns. If any points remain unclear, or if additional questions have come up after reading our replies, we would be happy to clarify them while the discussion phase is still open.
>
> We sincerely appreciate your time and consideration.

---

### Author Response · Authors · 2025-11-21
**Summary of reviews and shared reponse**

We thank all reviewers for their thoughtful and constructive feedback. Overall, our paper is described as **relevant, well-written, and easy to follow**, with an average presentation score of 3.5. The methodological contribution, in particular the NF-based action-inversion reward-sharing trick, was consistently recognized as **novel and principled**. The empirical evaluation was characterized as **comprehensive and convincing**.

# Updated manuscript

We have uploaded a revised version of the manuscript incorporating feedback from all reviewers; changes are highlighted in red. In summary, we've included and additional baseline as suggested by reviewer fyEH. We've added two new sections on "Limitations and Discussions" and "Future Work", as well as clarifying statements and minor revisions throughout the manuscript. We've added a related works section on "Learning from Demonstrations", based on the points made by reviewer TwB7. As suggest by reviewer add8, we've added a sensitivity analysis on the arbitrator's Boltzmann temperature as well as an analysis on the number of priors used by APC under Appendix F.

We strongly believe these changes have improved the manuscript and again thank all reviewers for their valuabel time and feedback.


# Regarding APC's computational overhead
A shared concern among reviewers was APC's computational overhead, to which we reply once and jointly below.

APC incurs high computational and memory overhead, scaling linearly with the number of actors. We agree that this is an apparent limitation of our approach, and that it was not sufficiently discussed in the originally submitted version of the manuscript. We have made this more transparent by explicitly discussing the computational overhead in the new “Discussion and Limitations” section of the revised manuscript.

With that being said, we have identified and are experimenting with several ways to **reducing APC's memory and computational demands**. These include maintaing a single, shared replay buffer: We can store the environment actions $\mathbf{a} \in \mathcal{A}$ and apply the inverse action trick to obtain latent actions $\mathbf{z}^{(i)} \in \mathcal{Z}^{(i)}$ only after sampling a training batch. Initial experiments with a unified, central critic architecture are promising an alleviate a large part of the computational load. In addition, only updating the actor selected at time $t$, instead of updating all actors, can also reduce the wall-clock training time. We experienced no impact on asymptotic performance, though slightly slower initial exploration, when applying this heuristic. We have added these improvements as ongoing future work in the revised manuscript.

Please see our individual responses regarding all other points.

---

### Author Response · Authors · 2025-12-03
**Final summary for AC**

Dear AC,

To facilitate the final decision, we offer the following concise summary:

Three of the four reviewers (add8, TwB7, LZVv) are in clear agreement that the paper is well-written, well-motivated, and technically sound, with strong and convincing experiments. They independently highlight the clarity of the method, the relevance of the problem, and the quality of the empirical evaluation.

The remaining review (feYH) is a clear and somewhat problematic outlier. The review seems to be based on a fundamental misunderstanding of the method (despite the reviewer's confidence score of 5), and several additional criticisms contradict the assessments of the other reviewers. We provide a more detailed explanation of these issues in our last response to that reviewer.

Thank you for your time, consideration, and effort given the challenging and unusual situation surrounding the leak.

---

### Meta-Review · Area_Chair_WELW · 2026-01-11

**Summary:**

The authors propose Adaptive Policy Composition (APC-RL): a hierarchical approach that learns multiple Normalizing Flow (NF) behavioral priors from different demonstration sources and trains corresponding prior-based low-level actors (plus a prior-free actor) with an off-policy RL algorithm (SAC). Across reviewers, there is broad agreement the paper is well motivated, clearly presented, and provides a meaningful algorithmic contribution (adaptive multi-prior selection + NF-based reward-sharing) that improves robustness under misaligned demonstrations—an important practical regime. After carefully reading the paper, review and author responses, the AC agrees with the majority of the reviewers on accepting the paper.

**Reviewer Concerns:**

see Summary

**Reviewer Scores:**

see Summary

---

### Decision · Program_Chairs · 2026-01-26

Accept (Poster)